# A Conditional Independence Test in the Presence of Discretization

**Boyang Sun**[1], **Yu Yao**[4], **Guang-Yuan Hao**[1], **Yumou Qiu**[3*], **Kun Zhang**[2,1*]

[1] Mohamed bin Zayed University of Artificial Intelligence
[2] Carnegie Mellon University
[3] Peking University
[4] The University of Sydney

## Abstract

Testing conditional independence (CI) has many important applications, such as Bayesian network learning and causal discovery. Although several approaches have been developed for inferring CI relationships among observed variables, these existing methods generally fail when the variables of interest cannot be directly observed and only discretized values of those variables are available. For example, if $X_1$, $\tilde{X}_2$ and $X_3$ are the observed variables, where $\tilde{X}_2$ is a discretization of the latent variable $X_2$, applying the existing methods to the observations of $X_1$, $\tilde{X}_2$ and $X_3$ would lead to a false conclusion about the underlying CI of variables $X_1$, $X_2$ and $X_3$. Motivated by this, we propose a CI test specifically designed to accommodate the presence of discretization. To achieve this, a bridge equation and nodewise regression are used to recover the precision coefficients reflecting the conditional dependence of the latent continuous variables under the nonparanormal model. We propose a test statistic and derive its asymptotic distribution under the null hypothesis of CI. Theoretical analysis, along with empirical validation on various datasets, rigorously demonstrates the effectiveness of our testing methods. Our code implementation can be found in `https://github.com/boyangaaaaa/DCT`.

## 1 Introduction

Independence and conditional independence (CI) are fundamental concepts in statistics. They are leveraged for exploring queries in statistical inference, such as sufficiency, parameter identification, and ancillarity (Dawid, 1979). They also play a central role in emerging areas such as causal discovery (Koller and Friedman, 2009), graphical model learning, and feature selection (Xing et al., 2001). Tests for CI have attracted increasing attention from both theoretical and application sides.

Formally, the problem is to test the CI of two variables $X_i$ and $X_j$ given a random vector (a set of other variables) $\boldsymbol{Z}$. In statistical notation, the null hypothesis is written as $H_0 : X_i \perp\!\!\!\perp X_j \mid \boldsymbol{Z}$, where $\perp\!\!\!\perp$ denotes "independent from". The alternative hypothesis is written as $H_1 : X_i \not\perp\!\!\!\perp X_j \mid \boldsymbol{Z}$, where $\not\perp\!\!\!\perp$ denotes "dependent with". The null hypothesis implies that once $\boldsymbol{Z}$ is known, the values of $X_i$ provide no additional information about $X_j$, and vice versa. Various tests have been designed to address different scenarios, including Gaussian variables with linear dependence (Yuan and Lin, 2007; Peterson et al., 2015; Mohan et al., 2012; Ren et al., 2015) and non-linear dependence (Fukumizu et al., 2004; Zhang et al., 2012; Strobl et al., 2019; Sen et al., 2017; Aliferis et al., 2010) (*For detailed related work, please refer to App. E*).

Given observations of $X_i$, $X_j$, and $\boldsymbol{Z}$, the CI relationship can be effectively tested with the existing methods. However, in many scenarios, accurately measuring continuous variables of interest is challenging due to limitations in data collection. Sometimes the data obtained are approximations represented as discretized values. For example, in finance, variables such as asset values cannot be measured and are binned into ranges for assessing investment risks (e.g., sell, hold, and strong buy) (Changsheng and Yongfeng, 2012; Damodaran, 2012). Similarly, in mental health, anxiety levels are

---

*Co-corresponding authors: Yumou Qiu (qiuyumou@math.pku.edu.cn), Kun Zhang(kunz1@cmu.edu)

often assessed using scales like the GAD-7, which categorizes responses into levels such as mild, moderate, or severe (Mossman et al., 2017; Johnson et al., 2019). In the entertainment industry, the quality of movies is typically summarized through viewer ratings (Sparling and Sen, 2011; Dooms et al., 2013).

When discretization is present, existing CI tests can fail to determine the CI relationships of the underlying variables. This issue arises because existing CI tests treat discretized observations as observations of continuous variables, leading to incorrect conclusions about their CI relationships. More precisely, the problem lies in the discretization process, which introduces new discrete variables. Consequently, *although the intent is to test the CI of the underlying continuous variables, what is being tested is the CI involving a mix of both continuous and newly introduced discrete variables*. In general, this CI relationship is inconsistent with the one among the underlying continuous variables.

As illustrated in Fig. 1, we show different data-generative processes using causal graphical models (Pearl, 2000) in the presence of discretization. A gray node indicates an observable variable, while a white node indicates a latent variable. Variables denoted by $X_j$ (without a tilde $\sim$) represent continuous variables, which may not be observed; while variables denoted by $\tilde{X}_j$ represent observed discretized variables derived from $X_j$ due to discretization. In Fig. 1(a), $X_2$ is latent, and only its discrete counterpart $\tilde{X}_2$ is observed. In this case, rather than observing $X_1$, $X_2$, and $X_3$, we only observe $X_1$, $\tilde{X}_2$, and $X_3$.

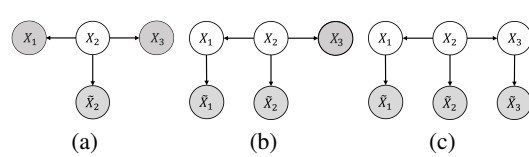

(a)  (b)  (c)

Figure 1: We illustrate data-generative processes with causal graphical models. The discretization process introduces new discrete variables indicated by a tilde ($\sim$).

Existing CI methods use these observations to test **whether** $X_1 \perp\!\!\!\perp X_3 \mid \{X_2\}$, but what is actually being tested is **whether** $X_1 \perp\!\!\!\perp X_3 \mid \{\tilde{X}_2\}$. In fact, according to the *causal Markov condition* (Spirtes et al., 2000), it can be inferred from Fig. 1(a) that $X_1 \perp\!\!\!\perp X_3 \mid \{X_2\}$ and $X_1 \not\perp\!\!\!\perp X_3 \mid \{\tilde{X}_2\}$. This mismatch leads to existing CI methods, that employ observations to check the CI relationships between $X_1$ and $X_3$ given $X_2$, to reach incorrect conclusions. Due to the same reason, checking the CI also fails in Fig 1(b) and Fig 1(c).

In this paper, we design a CI test specifically for handling the presence of discretization. An appropriate test statistic for the CI of latent continuous variables, based solely on discretized observations, is derived. To develop this test, we first estimate the covariance between latent continuous variables and discretized observations. This is achieved by constructing bridge equations that enable the estimation of covariance using statistics derived from discretized observations. Subsequently, to utilize the estimated covariance of latent continuous variables for testing CI relationships, we apply a node-wise regression approach (Callot et al., 2019), which allows us to derive test statistics for CI based on the estimated covariance. By assuming that the continuous variables follow a Gaussian distribution, we can derive the asymptotic distribution of the test statistics under the null hypothesis of CI. Our major contributions include:

- We develop a CI test for ensuring accurate analysis in scenarios where data has been discretized, which are common due to limitations in data collection or measurement techniques.
- Our CI test can handle various scenarios including 1). Both variables $X_i$ and $X_j$ are discretized 2). Both variables $X_i$ and $X_j$ are continuous. 3). One of the variables $X_i$ or $X_j$ is discretized.
- We compare our test with the existing methods on both synthetic and real-world datasets, confirming that our method can effectively estimate the CI of the underlying continuous variables and outperform the existing tests applied on the discretized observations.

## 2 PROBLEM SETTING AND NEED FOR CORRECTION

**Problem Setting** Consider a set of independent and identically distributed (i.i.d.) $p$-dimensional random vectors, denoted as $\tilde{\boldsymbol{X}} = [X_1, X_2, \ldots, \tilde{X}_j, \ldots, \tilde{X}_p]$. In this set, some variables, indicated by a tilde ($\sim$), such as $\tilde{X}_j$, follow a discrete distribution. For each such variable, there exists a corresponding latent Gaussian random variable $X_j$. The transformation from $X_j$ to $\tilde{X}_j$ is governed

by an unknown monotone nonlinear function $g_j$ and a thresholding function $f_j$. The function $f_j \circ g_j : \mathcal{X} \to \tilde{\mathcal{X}}$ maps the continuous domain of $X_j$ onto the discrete domain of $\tilde{\mathcal{X}}_j$. Specifically, for each variable $X_j$, there exists a finite constant vector $\mathbf{d}_j = [d_{j,1}, \ldots, d_{j,M-1}]$ characterized by increasing elements such that

$$\tilde{X}_j = f_j(g_j(X_j)) = \begin{cases} 1 & g_j(X_j) < d_{j,1} \\ m & d_{j,m-1} < g_j(X_j) < d_{j,m} \\ M & g_j(X_j) > d_{j,M-1} \end{cases} \tag{1}$$

This model is also known as the nonparanormal model (Liu et al., 2009). The cardinality of the domain after discretization is at least 2 and smaller than infinity. Our goal is to assess both conditional and unconditional independence among the variables of the vector $\boldsymbol{X} = [X_1, X_2, \ldots, X_p]$. In our model, we assume $\boldsymbol{X} \sim N(0, \boldsymbol{\Sigma})$, $\boldsymbol{\Sigma}$ only contain 1 among its diagonal, i.e., $\sigma_{j,j} = 1$ for all $j \in [1, \ldots, p]$. One should note this assumption is *without loss of generality*. We provide a detailed discussion of our assumption in App. B.9.

**Why the correction is needed?** We aim to propose a CI test that serves as a correction to infer the correct CI relationships among the latent continuous variables of interest. One question that arises is whether the discretized variables exhibit the same conditional independence as their original continuous counterparts, i.e., the correction is not needed. This concern becomes more significant when the level of discretization is high. To show the effect of discretization, we present the following theorem, using Gaussian random variables as an example, to demonstrate that discretization inevitably introduces distortions. These distortions can lead to incorrect conclusions about CI relationships. The proof can be found in Appendix B.1.

**Theorem 2.1.** *Let $X_1, X_2$ and $X_3$ be jointly Gaussian random variables that are mutually dependent, such that $X_1 \perp\!\!\!\perp X_3 | X_2$, $\tilde{X}_2 = f_j(g_j(X_2))$ is the discretized observation as defined in equation 1. Then the conditional independence between $X_1$ and $X_3$ given $\tilde{X}_2$ doesn't hold, i.e., $X_1 \not\perp\!\!\!\perp X_3 | \tilde{X}_2$.*

## 3 DCT: A DISCRETIZATION-AWARE CI TEST

**Notation** Throughout this work, we use $X_j$ to denote the $j$-th component of the vector of variables $\boldsymbol{X}$. We denote the sample mean of $X_j$ by $\mathbb{E}_n[X_j]$, and the expectation by $\mathbb{E}[X_j]$. The empirical probability is represented by $\mathbb{P}_n$ whereas the true probability is denoted by $\mathbb{P}$. For a matrix $\mathbf{X}$, $\mathbf{X}_{-j}$ represents all columns of $\mathbf{X}$ except the $j$-th column, $\mathbf{X}_{-j,-j}$ denotes the submatrix obtained by removing both the $j$-th column and row, and $\mathbf{X}_{-j,j}$ represents the $j$-th column of $\mathbf{X}$ with the $j$-th row removed. For any parameter $\alpha$, we use $\hat{\alpha}$ to denote its estimation. $\mathbb{1}\{\text{condition}\}$ is 1 if the condition holds true, 0 otherwise. For a full notation table, please refer to App. A.

To develop a CI test, we need to design a test statistic that can reflect the conditional dependence relation and be computable using observations only. Next, it is essential to derive the underlying distribution of this statistic under the null hypothesis that the tested variables are conditionally (or unconditionally) independent. By calculating the value of the test statistic and assessing if this statistic is likely to be drawn from the derived distribution (i.e., calculating the *p-value* and comparing it with the significance level $\alpha$), we can decide if the null hypothesis should be rejected.

Our objective is to deduce the independence and CI relationships within the original multivariate Gaussian variable $\boldsymbol{X}$, based on its discretized observations $\tilde{\boldsymbol{X}}$. In the context of a multivariate Gaussian model, this challenge is directly equivalent to constructing statistical inferences for its covariance matrix $\boldsymbol{\Sigma} = (\sigma_{i,j})$ and its precision matrix $\boldsymbol{\Omega} = (\omega_{j,k}) = \boldsymbol{\Sigma}^{-1}$ (Baba et al., 2004). The covariance matrix $\boldsymbol{\Sigma}$ captures the pairwise covariances, while the precision matrix $\boldsymbol{\Omega}$ encodes CI relationships. Specifically, the entry $\omega_{j,k}$ represents the partial correlation coefficient between variables $X_j$ and $X_k$, which determines their CI given other variables. Technically, we are interested in two things: (1) the calculation of the covariance $\hat{\sigma}_{i,j}$ and the precision coefficient (or the partial correlation coefficient) $\hat{\omega}_{j,k}$, serving as the estimation of $\sigma_{i,j}$ and $\omega_{j,k}$ respectively; (2) the derivation of the distribution of $\hat{\sigma}_{i,j} - \sigma_{i,j}$ and $\hat{\omega}_{j,k} - \omega_{j,k}$ under the null hypothesis of independence and CI.

In the rest of this section, we discuss three key components: (1) we introduce **bridge equations** to estimate the covariance $\sigma_{i,j}$; (2) we derive the distribution of $\hat{\sigma}_{i,j} - \sigma_{i,j}$, showing it to be **asymptotically normal**; and (3) we use **nodewise regression** to establish the relationship between

the covariance matrix $\mathbf{\Sigma}$ and the precision matrix $\mathbf{\Omega}$. We show that the regression parameter $\beta_{j,k}$ serves as a proxy for the precision matrix entry $\omega_{j,k}$. Leveraging the distribution of $\hat{\sigma}_{i,j} - \sigma_{i,j}$, we demonstrate that $\hat{\beta}_{j,k} - \beta_{j,k}$ is also **asymptotically normal**.

## 3.1 Estimating Covariance Through Observations

Our first task is to establish the connection between the underlying covariance $\sigma_{i,j}$ of the continuous pair $X_i$ and $X_j$ with their observed counterparts. Due to discretization, the sample covariance matrix computed from $\tilde{\mathbf{X}}$ is inconsistent with the covariance matrix of $\mathbf{X}$. To obtain the estimation $\hat{\sigma}_{i,j}$ consistent with $\sigma_{i,j}$, the bridge equation is leveraged. In general, it takes the form:

$$\hat{\tau}_{i,j} = T(\hat{\sigma}_{i,j}; \hat{\mathbf{\Lambda}}), \tag{2}$$

where $\hat{\sigma}_{i,j}$ is the estimated covariance, $\hat{\tau}_{i,j}$ is a statistic that can also be estimated from observations, and $\hat{\mathbf{\Lambda}}$ is a set of additional parameters required by the function $T(\cdot)$. The specific form of the function $T(\cdot)$ will be derived later. Both $\hat{\tau}_{i,j}$ and $\hat{\mathbf{\Lambda}}$ should be able to be calculated purely relying on observations. *Then, given the calculated $\hat{\tau}_{i,j}$ and $\hat{\mathbf{\Lambda}}$, $\hat{\sigma}_{i,j}$ can be obtained by solving the bridge equation.* As a result, the covariance matrix $\mathbf{\Sigma}$ of $\mathbf{X}$ can be estimated, which contains information about both unconditional independence and CI (which can be derived from its inverse).

To estimate the covariance of a latent multivariate Gaussian distribution, we need to design $\hat{\tau}_{i,j}$, $\hat{\mathbf{\Lambda}}$, and $T(\cdot)$. Notably, bridge equations have to be designed to handle the possible cases: C1. both observed variables are discretized; C2. one variable is continuous while the other is discretized. For C3. both variables remain continuous, we can easily take its sample covariance as the estimated covariance. We will show that cases C1 and C2 can be merged into a single form of bridge equation with different parameters and a binarization operation applied to the observations. Our bridge equations are presented in Def. 3.1, Def. 3.2.

### 3.1.1 Bridge Equations for Discretized and Mixed Pairs

Let us first address the challenging cases where both observed variables are discretized or where one variable is continuous while the other is discretized. In general, different bridge equations would need to be designed to handle each case individually. *However, in our analysis, we provide a unified bridge equation that applies to both cases.* This is achieved by *binarizing* the observed variables, thereby unifying both cases into a binary case. As some information may be lost in the binarization process, this unification may require more data samples compared to using tailored bridge functions for each specific case. Improving sample efficiency with tailored bridge equations is left for future work.

Theoretically, continuous variables and discrete variables can be further discretized into binary variables. Imagine we have the observed variable $\tilde{X}_i$ with the possible values "low", "medium", "high", we can create a dividing point: everything above becomes "very high", everything below becomes "very low". This binarization process is also applicable to the continuous variable. Note that $\tilde{X}_j$ is just the discretized version of its corresponding continuous variable $X_j$, this dividing point directly responds to a specific value in the original continuous domain, which we denote as the boundary $h_j$. Multiple choices of $h_j$ are possible. In this paper, we define $h_j$ as the boundary in the continuous domain that corresponds to the mean of its discretized counterpart $\tilde{X}_j$. Mathematically, we define $h_j$ as follows: for any single discretized variable $\tilde{X}_j$, there exists a constant $c_j$ such that $h_j = g_j^{-1}(c_j)$ satisfying

$$\mathbb{1}\{\tilde{x}_j^l > \mathbb{E}[\tilde{X}_j]\} = \mathbb{1}\{g_j(x_j^l) > c_j\} = \mathbb{1}\{x_j^l > h_j\},$$

where $\tilde{x}_j^l$ is the $j$-th sample of $\tilde{X}_j$, and $x_j$ is the $j$-th sample of $X_j$.

**Estimating the boundary** Since the continuous variable $X_j$ follows a normal distribution according to our assumption, we can thus construct the relation $\mathbb{P}(\tilde{X}_j > \mathbb{E}[\tilde{X}_j]) = 1 - \Phi(h_j)$, where $\Phi$ is the cumulative distribution function (cdf) of a standard normal distribution. Although we do not have access to the true probability, we can easily obtain its estimation by counting how many samples drop in the region larger than its sample mean. Specifically,

$$\hat{h}_j = \Phi^{-1}(1 - \hat{\tau}_j), \tag{3}$$

where $\hat{\tau}_j = \frac{1}{n} \sum_{l=1}^n \mathbb{1}\{\tilde{x}_j^l > \mathbb{E}_n[\tilde{X}_j]\}$, serving as the estimation of $\mathbb{P}(\tilde{X}_j > \mathbb{E}[\tilde{X}_j])$. We further denote $\bar{\Phi}(\cdot) = 1 - \Phi(\cdot)$.

**Intuition of estimating covariance** The question now is to estimate the latent covariance $\sigma_{i,j}$ for the observed discrete pair $(\tilde{X}_i, \tilde{X}_j)$ or mixed pair $(\tilde{X}_i, X_j)$. Leveraging the binarization process, there exists boundaries $h_i$, $h_j$ that partition the continuous variables pair $X_i$ and $X_j$ to a $2 \times 2$ contingency table. The area of each cell in this table represents the joint probability of the pair $(X_i, X_j)$ falling with a specific region defined by those boundaries. In this paper, we focus on the top-right cell of the contingency table, which represents the joint probability of both variables exceeding their respective boundaries.

Let $Z_1$ and $Z_2$ denote random variables. Mathematically, we denote $\bar{\Phi}(z_1, z_2; \rho) = \mathbb{P}(Z_1 > z_1, Z_2 > z_2)$, where $(Z_1, Z_2)$ follows a bivariate normal distribution with mean zero, variance one and covariance $\rho$. For a discretized pair of observed variables $(\tilde{X}_i, \tilde{X}_j)$, we define

$$\tau_{i,j} := \mathbb{P}(\tilde{X}_i > \mathbb{E}[\tilde{X}_i], \tilde{X}_j > \mathbb{E}[\tilde{X}_j]) = \bar{\Phi}(h_i, h_j; \sigma_{i,j}).$$

The above equation shows that the probability of discretized variables larger than their mean is a function of underlying covariance. It serves as a key to estimating the covariance. The probability in the above equation can be estimated by counting samples dropped into the region of both variables exceeding their sample means as follows:

$$\hat{\tau}_{i,j} := \mathbb{P}_n(\tilde{X}_i > \mathbb{E}_n[\tilde{X}_i], \tilde{X}_j > \mathbb{E}_n[\tilde{X}_j]) = \frac{1}{n} \sum_{l=1}^n \mathbb{1}\{\tilde{x}_i^l > \mathbb{E}_n[\tilde{X}_i], \tilde{x}_j^l > \mathbb{E}_n[\tilde{X}_j]\}. \tag{4}$$

Since $\bar{\Phi}(h_i, h_j; \sigma_{i,j})$ is a function of $\sigma_{i,j}$, by substituting the parameters $\tau_{i,j}, h_i, h_j$ as their estimation, we can construct the bridge equation as follows:

**Definition 3.1** (Bridge Equation for A Discretized-Variable Pair). For discretized variables $\tilde{X}_i$ and $\tilde{X}_j$, the bridge equation is defined as:

$$\hat{\tau}_{i,j} = T(\hat{\sigma}_{i,j}; \{\hat{h}_i, \hat{h}_j\}),$$

where $T(\hat{\sigma}_{i,j}; \{\hat{h}_i, \hat{h}_j\}) = \int_{z_1 > \hat{h}_i} \int_{z_2 > \hat{h}_j} \phi(z_1, z_2; \hat{\sigma}_{i,j}) \, dz_1 \, dz_2$, and $\phi$ is the probability density function of a bivariate normal distribution with mean zero and covariance $\hat{\sigma}_{i,j}$, we note that $\hat{h}_i, \hat{h}_j$ can be simply calculated using equation 3 and $\hat{\tau}_{i,j}$ can be calculated using equation 4.

Following the same idea, we can apply the same bridge equation to estimate the covariance of mixed pairs. The only difference is there is no need to estimate the boundary $\hat{h}_j$ for the continuous variable. Instead, we can incorporate its true mean of zero into the equation.

**Definition 3.2** (Bridge Equation for A Continuous-Discretized-Variable Pair). For one continuous variable $X_i$ and one discretized variable $\tilde{X}_j$, the bridge function is defined as:

$$\hat{\tau}_{i,j} = \mathbb{P}_n(X_i > 0, \tilde{X}_j > \mathbb{E}_n[\tilde{X}_j]) := \frac{1}{n} \sum_{l=1}^n \mathbb{1}\{x_i^l > 0, \tilde{x}_j^l > \mathbb{E}_n[\tilde{X}_j]\} = T(\sigma_{i,j}; \{0, \hat{h}_j\}),$$

and the function $T(\cdot)$ has the same form of Def. 3.1.

### 3.1.2 CALCULATION OF ESTIMATED COVARIANCE

For the continuous case where there is no discretization transformation, the sample covariance provides a consistent estimation of the true one. That is, for an observable pair of continuous variables $(X_i, X_j)$, we can simply obtain the analytic solution of estimated covariance:

$$\hat{\sigma}_{i,j} = \frac{1}{n} \sum_{l=1}^n x_i^l x_j^l - \frac{1}{n} \sum_{l=1}^n x_i^l \frac{1}{n} \sum_{l=1}^n x_j^l \tag{5}$$

For the cases involving the discretized variable as proposed in Def. 3.1 and Def. 3.2, we can rely on the property that variance $\Sigma$ only contains 1 among the diagonal, which implies the covariance $\sigma_{i,j}$ should vary from $-1$ to $1$. Thus, we can calculate the estimated covariance by solving the objective

$$\hat{\sigma}_{i,j} = \arg\min_{\sigma'_{i,j}} ||\hat{\tau}_{i,j} - T(\sigma'_{i,j}; \{\hat{h}_i, \hat{h}_j\})||^2 \quad s.t. -1 < \sigma'_{i,j} < 1. \tag{6}$$

The $\hat{\tau}_{i,j}$ is a one-to-one mapping with calculated $\hat{\sigma}_{i,j}$ given $\hat{h}_i$ and $\hat{h}_j$, which is proved in App. B.3

## 3.2 UNCONDITIONAL INDEPENDENCE TEST

The estimation of covariance $\hat{\sigma}_{i,j}$ can be effectively solved using the designed bridge equation. Now, we focus on deriving the distribution of $\hat{\sigma}_{i,j} - \sigma_{i,j}$. These results are used as an unconditional independence test in the presence of discretization. Moreover, Thm. 3.3, Lem. 3.4, Lem. 3.5 and Lem. 3.6 will be leveraged in the derivation process of the CI test in Section 3.3. The detailed derivation steps for both the unconditional independence test and the CI test are relatively complicated, therefore, we will provide a general intuition. For a complete derivation, please refer to the App. B.4.

Assume we are interested in the true parameter $\boldsymbol{\theta}_0$, e.g., for discretized pairs, $\boldsymbol{\theta}_0 = (\sigma_{i,j}, h_i, h_j)$. We denote $\hat{\boldsymbol{\theta}}$ as its estimation which is close to $\boldsymbol{\theta}_0$, and $f(\boldsymbol{\theta})$ is a continuous function. By leveraging Taylor expansion, we have

$$f(\hat{\boldsymbol{\theta}}) = f(\boldsymbol{\theta}_0) + f'(\boldsymbol{\theta}_0)(\hat{\boldsymbol{\theta}} - \boldsymbol{\theta}_0) + \ldots, \tag{7}$$

where the second-order terms and more are omitted, which directly constructs the relationship between the estimated parameter with the true one. Rearrange the term, we get $\hat{\boldsymbol{\theta}} - \boldsymbol{\theta}_0 = (f(\hat{\boldsymbol{\theta}}) - f(\boldsymbol{\theta}_0))/f'(\boldsymbol{\theta}_0)$. If the denominator is a constant and the numerator can be expressed as a sum of i.i.d samples, we can see $\hat{\boldsymbol{\theta}} - \boldsymbol{\theta}_0$ will be asymptotically normal (Van der Vaart, 2000).

Let $\boldsymbol{\psi}_{\hat{\boldsymbol{\theta}}} = [f_{\hat{\boldsymbol{\theta}}}^1(\cdot), \ldots]^T$ contains a group of functions parameterized by $\hat{\boldsymbol{\theta}}$. We define the functions evaluated at one sample as $\boldsymbol{\psi}_{\hat{\boldsymbol{\theta}}}^l = \boldsymbol{\psi}_{\hat{\boldsymbol{\theta}}}(\mathbf{z}^l)$, where $\mathbf{z}^l$ denotes the $l$-th sample point. We define the sample mean of these functions evaluated at $n$ points as $\mathbb{E}_n[\boldsymbol{\psi}_{\hat{\boldsymbol{\theta}}}] = \frac{1}{n}\sum_{l=1}^n \boldsymbol{\psi}_{\hat{\boldsymbol{\theta}}}^l$, similarly, $\mathbb{E}_n[\boldsymbol{\psi}_{\hat{\boldsymbol{\theta}}}\boldsymbol{\psi}_{\hat{\boldsymbol{\theta}}}^T] = \frac{1}{n}\sum_{l=1}^n \boldsymbol{\psi}_{\hat{\boldsymbol{\theta}}}^l {\boldsymbol{\psi}_{\hat{\boldsymbol{\theta}}}^l}^T$ and $\boldsymbol{\psi}_{\hat{\boldsymbol{\theta}}}'$ denotes the Jacobian matrix $\frac{\partial \boldsymbol{\psi}_{\hat{\boldsymbol{\theta}}}}{\partial \hat{\boldsymbol{\theta}}}$. We now provide the main result of derived distribution $\hat{\sigma}_{i,j} - \sigma_{i,j}$ under the hull hypothesis that tested pairs are independent.

**Theorem 3.3** (Independence Test). *Under the null hypothesis that the Gaussian variables $(X_i, X_j)$ are statistically independent $\sigma_{i,j} = 0$, the test statistics $\hat{\sigma}_{i,j}$ obtained according to Def. 3.1 for discretized pairs $(\tilde{X}_i, \tilde{X}_j)$, Def. 3.2 for mixed pairs $(X_i, \tilde{X}_j)$ and equation 5 for continuous pairs, is asymptotically normal:*

$$\sqrt{n}(\hat{\sigma}_{i,j} - \sigma_{i,j}) \xrightarrow{d} N\left(0, ((\mathbb{E}_n[\boldsymbol{\psi}_{\hat{\boldsymbol{\theta}}}'])^{-1}\mathbb{E}_n[\boldsymbol{\psi}_{\hat{\boldsymbol{\theta}}}\boldsymbol{\psi}_{\hat{\boldsymbol{\theta}}}^T](\mathbb{E}_n[\boldsymbol{\psi}_{\hat{\boldsymbol{\theta}}}'^T])^{-1})_{1,1}\right), \tag{8}$$

*where the specific form of $\boldsymbol{\psi}_{\hat{\boldsymbol{\theta}}}^l$ are presented in Lem. 3.4, Lem. 3.5 and Lem. 3.6.*

We now provide the specific forms of $\boldsymbol{\psi}_{\hat{\boldsymbol{\theta}}}^l$. Since the variables being tested for independence can be both discretized, only one being discretized, or neither being discretized—-the form of $\boldsymbol{\psi}_{\hat{\boldsymbol{\theta}}}$ varies accordingly. The specific forms of $\boldsymbol{\psi}_{\hat{\boldsymbol{\theta}}}$ in these scenarios are defined as follows:

**Lemma 3.4.** *($\boldsymbol{\psi}_{\hat{\boldsymbol{\theta}}}^l$ for A Continuous-Variable Pair). For two continuous variables $X_i$ and $X_j$, where $\hat{\boldsymbol{\theta}} = \hat{\sigma}_{i,j}$, and their corresponding $l$-th samples $x_i^l, x_j^l$:*

$$\boldsymbol{\psi}_{\hat{\boldsymbol{\theta}}}^l := x_i^l x_j^l - \mathbb{E}_n[X_i]\mathbb{E}_n[X_j] - \hat{\sigma}_{i,j},$$

**Lemma 3.5** *($\boldsymbol{\psi}_{\hat{\boldsymbol{\theta}}}^l$ for A Discretized-Variable Pair). For discretized variables $\tilde{X}_i$ and $\tilde{X}_j$, where $\hat{\boldsymbol{\theta}} = (\hat{\sigma}_{i,j}, \hat{h}_i, \hat{h}_j)$, and their corresponding $l$-th samples $\tilde{x}_i^l, \tilde{x}_j^l$:*

$$\boldsymbol{\psi}_{\hat{\boldsymbol{\theta}}}^l := \begin{pmatrix} \hat{\tau}_{i,j}^l - T(\hat{\sigma}_{i,j}; \{\hat{h}_i, \hat{h}_j\}) \\ \hat{\tau}_i^l - \bar{\bar{\Phi}}(\hat{h}_i) \\ \hat{\tau}_j^l - \bar{\bar{\Phi}}(\hat{h}_j) \end{pmatrix},$$

*where $\hat{\tau}_{i,j}^l = \mathbb{1}\{\tilde{x}_i^l > \mathbb{E}_n[\tilde{X}_i], \tilde{x}_j^l > \mathbb{E}_n[\tilde{X}_j]\}, \hat{\tau}_i^l = \mathbb{1}\{\tilde{x}_i^l > \mathbb{E}_n[\tilde{X}_i]\}$, and similarly for $\hat{\tau}_j^l$.*

**Lemma 3.6** *($\boldsymbol{\psi}_{\hat{\boldsymbol{\theta}}}^l$ for A Continuous-Discretized-Variable Pair). For one discretized variable $\tilde{X}_j$ and one continuous variable $X_i$, where $\hat{\boldsymbol{\theta}} = (\hat{\sigma}_{i,j}, \hat{h}_j)$, and their corresponding $l$-th sample point $\tilde{x}_j^l, x_i^l$:*

$$\boldsymbol{\psi}_{\hat{\boldsymbol{\theta}}}^l := \begin{pmatrix} \hat{\tau}_{i,j}^l - T(\hat{\sigma}_{i,j}; \{0, \hat{h}_j\}) \\ \hat{\tau}_j^l - \bar{\bar{\Phi}}(\hat{h}_j) \end{pmatrix},$$

*where $\hat{\tau}_{i,j}^l = \mathbb{1}\{x_i^l > 0, \tilde{x}_j^l > \mathbb{E}_n[\tilde{X}_j]\}, \hat{\tau}_j^l = \mathbb{1}\{\tilde{x}_j^l > \mathbb{E}_n[\tilde{X}_j]\}$.*

Derivation of forms of $\psi_{\hat{\theta}}$ for different cases and their corresponding distribution defined in Eq equation 8 can be found in App. B.5, App. B.6, App. B.7. Up to this point, our discussion has been confined to the case of covariance $\sigma_{i,j}$, the indicator of unconditional independence. In the next section, we will present the results of our CI test.

## 3.3 CONDITIONAL INDEPENDENCE (CI) TEST

To construct a CI test of our model, we are interested in two matters: calculation of the estimated precision coefficient $\hat{\omega}_{j,k}$ and the derivation of the corresponding distribution $\hat{\omega}_{j,k} - \omega_{j,k}$. While obtaining $\hat{\omega}_{j,k}$ from the $\hat{\Sigma}$ is straightforward, it leaves the inference problem unresolved. Thus, we leverage nodewise regression and show the regression parameter $\beta_{j,k}$ serving as a surrogate of testing for $\omega_{j,k} = 0$, we then construct the formulation of $\hat{\beta}_{j,k} - \beta_{j,k}$ as the combination of formulation of $\hat{\sigma}_{i,j} - \sigma_{i,j}$ and show it will also be asymptotically normal.

The following lemma formalizes the properties of nodewise regression that enable this approach:

**Lemma 3.7.** *[Nodewise Regression Properties] For a p-dimensional multivariate normal variable* $\boldsymbol{X} = (X_1, \ldots, X_p) \sim N(0, \boldsymbol{\Sigma})$ *with covariance matrix* $\boldsymbol{\Sigma}$ *and precision matrix* $\boldsymbol{\Omega} = \boldsymbol{\Sigma}^{-1} = (\omega_{j,k})_{1 \leq j,k \leq p}$. *For any* $j \in \{1, \ldots, p\}$, *consider the nodewise regression where each* $X_j$ *is regressed on all other variables:*

$$X_j = \sum_{k \neq j} X_k \beta_{j,k} + \epsilon_j,$$

*where* $\beta_{j,k}$ *is the regression coefficient of* $X_k$ *in predicting* $X_j$, $\boldsymbol{\beta}_j = (\beta_{j,k})_{k \neq j} \in \mathbb{R}^{p-1}$ *is the vector of all coefficients, and* $\epsilon_j$ *is the residual term. Then the following relationships hold:*

$$\boldsymbol{\beta}_j = \boldsymbol{\Sigma}_{-j,-j}^{-1} \boldsymbol{\Sigma}_{-j,j} \in \mathbb{R}^{p-1},$$
$$\beta_{j,k} = -\frac{\omega_{j,k}}{\omega_{j,j}}, \quad j \neq k. \tag{9}$$

The derivation can be found in App. B.8.1. The lemma establishes the deterministic relationships between the regression coefficient $\beta_{j,k}$ and the entry of precision matrix $\omega_{j,k}$. Since $\omega_{j,j}$ will never be zero (due to the positive definiteness $\boldsymbol{\Omega}$), we can conclude $\beta_{j,k}$ serves as an effective surrogate of $\omega_{j,k}$. Moreover, $\boldsymbol{\beta}_j$ can be expressed in terms of the submatrices of the covariance matrix $\boldsymbol{\Sigma}$. We can further conduct its estimation $\hat{\boldsymbol{\beta}}_j = (\hat{\beta}_{j,k})_{k \neq j} = \hat{\boldsymbol{\Sigma}}_{-j,-j}^{-1} \hat{\boldsymbol{\Sigma}}_{-j,j}$, where the estimated covariance terms can be obtained using Def. 3.1, 3.2 and equation 5.

**Statistical Inference for** $\beta_{j,k}$    Nodewise regression offers a direct solution for the estimation problem. A pertinent inquiry pertains to the construction of the distribution of $\hat{\boldsymbol{\beta}}_j - \boldsymbol{\beta}_j$. It is crucial to recognize that the distribution of $\hat{\sigma}_{i,j} - \sigma_{i,j}$ is already established. Therefore, if we can conceptualize $\hat{\boldsymbol{\beta}}_j - \boldsymbol{\beta}_j$ as a linear combination of $\hat{\sigma}_{i,j} - \sigma_{i,j}$, the problem is directly solved, i.e., the $\hat{\boldsymbol{\beta}}_j - \boldsymbol{\beta}_j$ is linear combination of dependent Gaussian variables. The underlying relationship between these variables is as follows:

$$\hat{\boldsymbol{\beta}}_j - \boldsymbol{\beta}_j = -\hat{\boldsymbol{\Sigma}}_{-j,-j}^{-1} \left( (\hat{\boldsymbol{\Sigma}}_{-j,-j} - \boldsymbol{\Sigma}_{-j,-j}) \boldsymbol{\beta}_j - (\hat{\boldsymbol{\Sigma}}_{-j,j} - \boldsymbol{\Sigma}_{-j,j}) \right).$$

The derivation is provided in App. B.8.2. For ease of notation, we further express the distribution of the difference between the estimated covariance and the true covariance as

$$\hat{\sigma}_{i,j} - \sigma_{i,j} = \frac{1}{n} \sum_{l=1}^{n} \xi_{i,j}^l. \tag{10}$$

The specific form of $\xi_{i,j}^l$ is given in App. B.5, B.6, B.7 for different cases, respectively. For notational convenience, we express $\hat{\boldsymbol{\Sigma}}_{-j,-j} - \boldsymbol{\Sigma}_{-j,-j} = \frac{1}{n} \sum_{l=1}^{n} \boldsymbol{\Xi}_{-j,-j}^l$ and $\hat{\boldsymbol{\Sigma}}_{-j,j} - \boldsymbol{\Sigma}_{-j,j} = \frac{1}{n} \sum_{l=1}^{n} \boldsymbol{\Xi}_{-j,j}^l$, where $\xi_{i,j}$ is the element of the matrix $\boldsymbol{\Xi}$ at the position indexed by $(i,j)$. We propose the statistic and its asymptotic distribution for the CI test in the following theorem.

**Theorem 3.8** (Conditional Independence test)**.** *Under the null hypothesis that Gaussian variables* $X_j$ *and* $X_k$ *are conditional statistically independent given all other variables* $\boldsymbol{X}_{-\{jk\}}$, *i.e.,* $\beta_{j,k} = 0$, *the testing statistic*

$$\hat{\beta}_{j,k} = (\hat{\boldsymbol{\Sigma}}_{-j,-j}^{-1} \hat{\boldsymbol{\Sigma}}_{-j,j})_{[k]}, \tag{11}$$

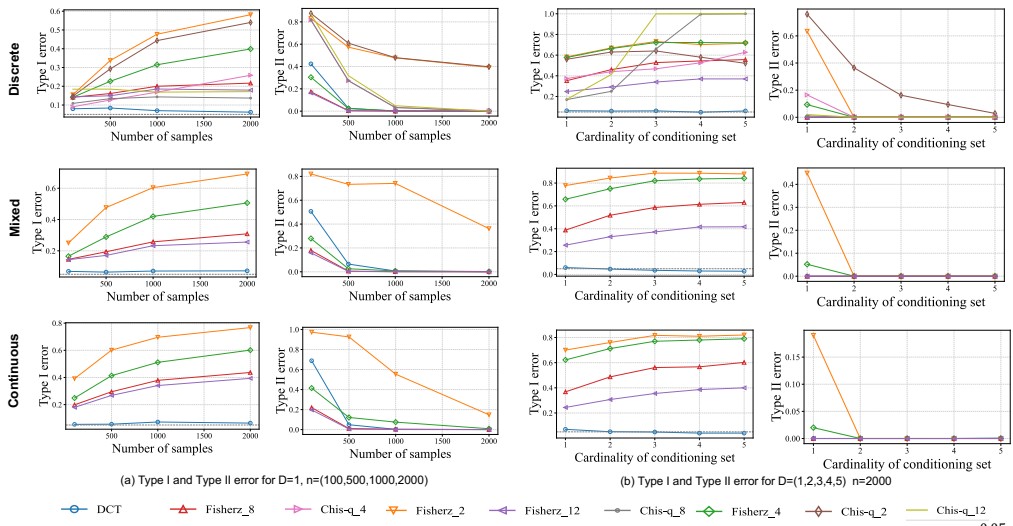

Figure 2: Comparison of results of Type I and calibrated Type II error (1 − power) for all three types of tested data (continuous, mixed, discrete) and different number of samples and cardinality of conditioning set. The suffix attached to a test's name denotes the cardinality of discretization; for example, "Fisherz_4" signifies the application of the Fisher-z test to data discretized into four levels. Chi-square test is only applicable for the discrete case.

*where $[k]$ denotes the element corresponding to the variable $X_k$ in $\hat{\boldsymbol{\Sigma}}_{-j,-j}^{-1}\hat{\boldsymbol{\Sigma}}_{-j,j}$, has the asymptotic distribution:*

$$\sqrt{n}(\hat{\beta}_{j,k} - \beta_{j,k}) \xrightarrow{d} N(0, \boldsymbol{a}^{[k]T}\frac{1}{n}\sum_{l=1}^{n} vec(\boldsymbol{B}_{-j}^{l})vec(\boldsymbol{B}_{-j}^{l})^{T}\boldsymbol{a}^{[k]}),$$

$$\text{where } \boldsymbol{B}^l = \begin{bmatrix} \boldsymbol{\Xi}_{-j,j}^{l}{}^{T} \\ \boldsymbol{\Xi}_{-j,-j}^{l} \end{bmatrix}, \quad \boldsymbol{a}^{[k]} = \begin{bmatrix} -(\hat{\boldsymbol{\Sigma}}_{-j,-j}^{-1})_{[k],:}^{T} \\ vec\left((\hat{\boldsymbol{\Sigma}}_{-j,-j}^{-1})_{[k],:}^{T}\tilde{\boldsymbol{\beta}}_{j}^{T}\right) \end{bmatrix},$$

*and $\tilde{\boldsymbol{\beta}}_j$ is $\boldsymbol{\beta}_j$ whose $\beta_{j,k} = 0$; vec is row-wise vectorization of a matrix, and $(\hat{\boldsymbol{\Sigma}}_{-j,-j}^{-1})_{[k],:}$ denotes the row in $\hat{\boldsymbol{\Sigma}}_{-j,-j}^{-1}$ that corresponds to $X_k$.*

In practice, we can plug in the estimation of regression parameter $\hat{\boldsymbol{\beta}}_j$ and set $\hat{\beta}_{j,k} = 0$ as the substitution of $\tilde{\boldsymbol{\beta}}_j$ to calculate the variance and do the CI test. Specifically, we can obtain the $\hat{\beta}_{j,k}$ using equation 11 where the estimated covariance terms can be calculated by solving the bridge equation Eq. 2. Under the null hypothesis that $\beta_{j,k} = 0$ (conditional independence), we can take the calculated $\hat{\beta}_{j,k}$ into the distribution defined in Thm. 3.8 and obtain the p-value. If the p-value is smaller than the predefined significance level $\alpha$ (normally set at 0.05), we will infer the tested pairs are conditionally dependent; otherwise, we do not. The detailed derivation of the Thm. 3.8 can be found in App. B.8.2. The pseudocode of DCT is provided in App. D.

## 4 EXPERIMENTS

We applied the proposed method DCT to synthetic data to evaluate its practical performance and compare it with Fisher-Z test (Fisher, 1921) (for all three data types) and Chi-Square test (F.R.S., 2009) (for discrete data only) as baselines. Specifically, we investigated its Type I and Type II error and its application in causal discovery. The experiments investigating its robustness, performance in denser graphs and effectiveness in a real-world dataset can be found in App. H.

### 4.1 ON THE EFFECT OF THE CARDINALITY OF CONDITIONING SET AND THE SAMPLE SIZE

Our experiment investigates the variations in Type I and Type II error (1 minus power) probabilities under two conditions. In the first scenario, we focus on the effects of modifying the sample size,

denoted as $n = (100, 500, 1000, 2000)$, while conditioning on a single variable. In the second, the sample size is held constant at 2000, and we vary the cardinality of the conditioning set, represented as $D = (1, 2, \ldots, 5)$. It is assumed that every variable within this conditioning set is effective, i.e., they influence the CI of the tested pairs. We repeat each test 1500 times.

We use $Y, W$ to denote the variables being tested and use $Z$ to denote the variables being conditioned on. The discretized versions of the variables are denoted with a tilde symbol (e.g., $\tilde{Z}$). For both conditions, we evaluate three distinct types of observations of tested variables: continuous observations for both variables $(Y, W)$, discrete observations for both variables $(\tilde{Y}, \tilde{W})$ and a mixed type $(\tilde{Y}, W)$. The variables in the conditioning set will always be discretized observations $(\tilde{Z})$.

To see how well the derived asymptotic null distribution approximates the true one, we verify if the probability of Type I error aligns with the significance level $\alpha$ preset in advance. We generate true continuous multivariate Gaussian data $Y, W$ from $Z_i$ (single $i = 1$ for the first scenario, and summed over $n$ for the second), structured as $a_i Z_i + E$ and $\sum_{i=1}^{n} a_i Z_i + E$, where $a_i$ is sampled from $U(0.5, 1.5)$ and $E$ follows a standard normal distribution, independent of all other variables. This ensures $Y \perp\!\!\!\perp W | Z$. The data are then discretized into $K = (2, 4, 8, 12)$ levels, with boundaries randomly set based on the variable range. The first column in Fig. 2 (a) (b) shows the resulting probability of Type I errors at the significance level $\alpha = 0.05$ compared with other methods.

A good test should have as small a probability of Type II error as possible, i.e., a larger power. To test the power of our DCT, we generate the continuous multivariate Gaussian data $Z_i$ from $Y, W$; constructed as $Z_i = a_i Y + b_i W + E$, where $a_i, b_i$ are sampled from $U(0.5, 1.5)$ and $E$ follows a standard normal distribution independent with all others, i.e., $Y \not\perp\!\!\!\perp W | Z$. The same discretization approach is applied here. One should note that directly comparing the p-value with a common predefined significance level is unfair since all baselines tend to produce very small p-values. Therefore, all tests are calibrated[1] in this experiment. The second column in Fig. 2 (a) and (b) correspondingly shows the calibrated Type II error as the number of samples and the cardinality of the conditioning set change, compared to other methods.

From Fig. 2 (a), we note that the Type I error rates with our derived null distribution are well-approximated at 0.05 across all three data types in both scenarios. In contrast, other testing methods show significantly higher Type I error rates, increasing with the number of samples and the size of the conditioning set. This indicates that such methods are more prone to erroneously concluding that tested variables are conditionally dependent. Additionally, while alternative tests demonstrate considerable power with smaller sample sizes, our approach requires a sample size of 1000 to achieve satisfactory power, particularly in mixed and continuous cases. A possible explanation for this phenomenon is that our method binarizes discretized data, which may not effectively utilize all observations. This aspect warrants further investigation in future research. Moreover, our test shows remarkable stability in response to changes in the number of conditioning sets.

## 4.2 APPLICATION IN CAUSAL DISCOVERY

Causal discovery aims at looking for the true causal structure from the data. Under the assumption of causal Markov condition that the causal relationships among variables can be expressed by a Directed Acyclic Graph (DAG) $\mathcal{G}$ and its statistical independence is entailed in this graphic model, faithfulness ensures that the statistical independencies observed in the data can be reliably used to infer the causal structure. Given both assumptions, constraint-based causal discovery, e.g., PC algorithm (Spirtes et al., 2000) recovers the graph structure relying on testing the conditional independence of observation. Apparently, in the presence of discretization, the failures of testing conditional independence will seriously impair the resulting DAG.

To evaluate the efficacy of the DCT, we construct the true DAG $\mathcal{G}$ utilizing the Bipartite Pairing (BP) model as detailed in (Asratian et al., 1998), with the number of edges being one fewer than the number of nodes. The subsequent generation of true multivariate Gaussian data involves assigning causal weights drawn from a uniform distribution $U \sim (0.5, 2)$ and incorporating noise via samples from a standard normal distribution for each variable. Following this, we binarize the data, setting the threshold randomly based on each variable's range. Our experiment is divided into two scenarios:

---

[1] Calibration is the process of empirically finding the decision threshold to match the desired significance level, ensuring accurate control of Type I error.

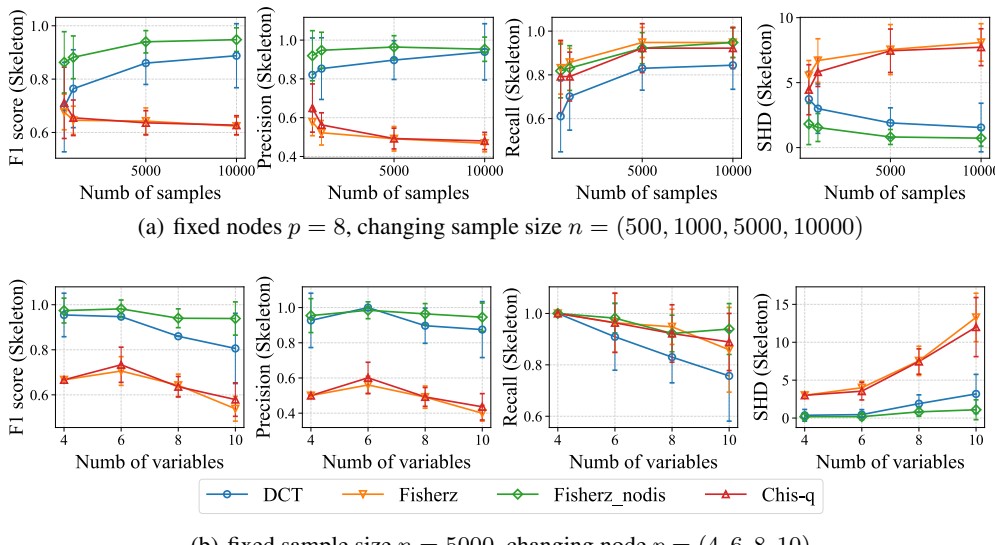

(a) fixed nodes $p = 8$, changing sample size $n = (500, 1000, 5000, 10000)$

(b) fixed sample size $n = 5000$, changing node $p = (4, 6, 8, 10)$

Figure 3: Experimental result of skeleton discovery on synthetic data for changing sample size (a) and changing number of nodes (b). Fisherz_nodis is the Fisher-z test applied to original continuous data. We evaluate $F_1$ (↑), Precision (↑), Recall (↑) and SHD (↓).

In the first, we set the number of samples $n = 5000$, with the number of nodes $p$ varying across 4, 6, 8, and 10. In the second scenario, we fix the number of nodes at $p = 8$ and explore sample sizes $n = (500, 1000, 5000, 10000)$.

A comparative analysis is performed using the PC algorithm integrated with various testing methods. Specifically, we compare DCT against the Fisher-z test applied to discretized data, the Chi-Square test, and the Fisher-z test on the original continuous data, the latter serving as a theoretical upper bound. Since the PC algorithm only returns a completed partially directed acyclic graph (CPDAG), we apply the same orientation rules from Dor and Tarsi (1992), as implemented by Causal-DAG (Chandler Squires, 2018), to convert a CPDAG into a DAG for easier comparison. We evaluate both the undirected skeleton and the directed graph using structural Hamming distance (SHD), F1 score, precision, and recall as evaluation metrics. For each setting, we run 10 graph instances with different seeds and report the mean and standard deviation for skeleton discovery in Fig. 3 and DAG discovery in Fig. 4 in App. C.

According to the result, DCT exhibits performance nearly on par with the theoretical upper bound across metrics such as F1 score, precision, and Structural Hamming Distance (SHD) when the number of variables ($p$) is small and the sample size ($n$) is large. DCT significantly outperforms both the Fisher-Z test and the Chi-square test. Notably, in almost all settings, the recall of DCT is lower than that of the baseline tests, which is reasonable *since these tests tend to infer conditional dependencies, thereby retaining all edges given the discretized observations.* For instance, a fully connected graph, would achieve a recall of 1.

## 5 CONCLUSION

In this paper, we present a new testing method tailored for scenarios commonly encountered in real-world applications, where variables, though inherently continuous, are only observable in their discretized forms. Our method distinguishes itself from existing CI tests by effectively mitigating the misjudgment introduced by discretization and accurately recovering the CI relationships of latent continuous variables. We substantiate our approach with theoretical results and empirical validation, underscoring the effectiveness of our testing methods.

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

*Appendix for*

# "A Conditional Independence Test in the Presence of Discretization"

Appendix organization:

# A  NOTATION TABLE

| Category | Description |
|---|---|
| **Number and Indices** | |
| $n$ | Number of samples |
| $p$ | Number of variables |
| $i, j, k$ | Index of a variable $i, j, k \in (1, \ldots, p)$ |
| $l$ | Index of a sample $l \in (1, \ldots, n)$ |
| **Random Variables** | |
| $\boldsymbol{X}$ | A vector of Gaussian variables |
| $\tilde{\boldsymbol{X}}$ | A vector of variables whose partial variables are discretized versions of $\boldsymbol{X}$ |
| $\boldsymbol{\Sigma}$ | Covariance of $\boldsymbol{X}$ |
| $\boldsymbol{\Sigma}_{-j,-j}$ | Submatrix of $\boldsymbol{\Sigma}$ with $j$-th row and $j$-th column removed |
| $\boldsymbol{\Sigma}_{-j,j}$ | $j$-th column of $\mathbf{X}$ with $j$-th row removed |
| $\boldsymbol{\Omega}$ | Precision matrix of $\boldsymbol{X}$, equals to $\boldsymbol{\Sigma}^{-1}$ |
| $X_j$ | $j$-th component of the $\boldsymbol{X}$ |
| $\boldsymbol{X}_{-\{j,k\}}$ | All other variables of $\boldsymbol{X}$ with $X_j$ and $X_k$ removed |
| $\sigma_{i,j}$ | Covariance between $X_i$ and $X_j$ |
| $\omega_{j,k}$ | Precision coefficient $\omega_{j,k}$ |
| $x_j^l$ | $l$-th sample of $X_j$ |
| $\tilde{x}_j^l$ | $l$-th sample of $\tilde{X}_j$ |
| $h_j$ | The boundary in the continuous domain that corresponds to the mean of $\tilde{X}_j$ |
| $\tau_j$ | Probability of $\tilde{X}_j$ larger than its mean: $\mathbb{P}(\tilde{X}_j > \mathbb{E}[\tilde{X}_j])$ |
| $\beta_{j,k}$ | Regression coefficient of $X_k$ in predicting $X_j$ |
| $\boldsymbol{\beta}_j$ | vector of all coefficients regressing $X_j$ |
| $\xi_{i,j}^l$ | Influence function component, it represents the influence of the $l$-th observation on the covariance estimation error |
| $\boldsymbol{\Xi}^l$ | Matrix form of $\xi^l$ |
| **Estimation of Variables** | |
| $\hat{\sigma}_{i,j}$ | Estimation of $\sigma_{i,j}$, calculated using equation 6, equation 5 |
| $\hat{\boldsymbol{\Sigma}}$ | Estimation of $\boldsymbol{\Sigma}$, matrix form of $\hat{\sigma}_{i,j}$ |
| $\hat{\omega}_{j,k}$ | Estimation of $\omega_{j,k}$ |
| $\hat{h}_j$ | Estimation of $h_j$, calculated using equation 3 |
| $\hat{\tau}_j$ | Estimation of $\tau_j$, calculated as $\frac{1}{n}\sum_{l=1}^{n} \mathbb{1}\{\tilde{x}_j^l > \mathbb{E}_n(\tilde{X}_j)\}$ |
| $\hat{\beta}_j$ | Estimation of $\hat{\beta}_j$, calculated as $\hat{\boldsymbol{\Sigma}}_{-j,-j}^{-1} \hat{\boldsymbol{\Sigma}}_{-j,j}$ |
| **Functions and Operators** | |
| $\mathbb{P}$ | True probability |
| $\mathbb{P}_n$ | Sample probability |
| $\mathbb{E}[Z]$ | Expectation of a random variable $Z$ |
| $\mathbb{E}_n[Z]$ | Sample mean of a random variable $Z$ over $n$ samples |
| $\mathbb{1}$ | 1 condition: is 1 if the condition is true, 0 otherwise |
| $\Phi(z)$ | Cumulative distribution function of a standard normal distribution |
| $\bar{\Phi}(z)$ | $1 - \Phi(z)$, corresponding to the $\mathbb{P}(Z > z)$ |
| $\bar{\Phi}(z_1, z_2; \rho)$ | $\mathbb{P}(Z_1 > z_1, Z_2 > z_2)$, where $(Z_1, Z_2)$ follows a bivariate normal distribution with mean zero, variance one and covariance $\rho$. |
| $\boldsymbol{\psi}_{\hat{\boldsymbol{\theta}}}$ | A group of functions parametrized by $\hat{\boldsymbol{\theta}}$ |
| $\boldsymbol{\psi}_{\hat{\boldsymbol{\theta}}}^l$ | $\boldsymbol{\psi}_{\hat{\boldsymbol{\theta}}}$ evaluated at sample $l$ |
| $\boldsymbol{\psi}_{\hat{\boldsymbol{\theta}}}'$ | Jacobian matrix of $\frac{\partial \boldsymbol{\psi}_{\hat{\boldsymbol{\theta}}}}{\partial \hat{\boldsymbol{\theta}}}$ |
| **For Discretized Pair $\tilde{X}_i, \tilde{X}_j$** | |
| $\tau_{i,j}$ | Probability of both $\tilde{X}_i$ and $\tilde{X}_j$ larger than their mean: $\mathbb{P}(\tilde{X}_i > \mathbb{E}[\tilde{X}_i], \tilde{X}_j > \mathbb{E}[\tilde{X}_j])$ |
| $\hat{\tau}_{i,j}$ | Estimation of $\tau_{i,j}$ : $\frac{1}{n}\sum_{l=1}^{n} \mathbb{1}\{\tilde{x}_i^l > \mathbb{E}_n[\tilde{X}_i], \tilde{x}_j^l > \mathbb{E}_n[\tilde{X}_j]\}$ |

| Category | Description |
|---|---|
| $\hat{\tau}_{i,j}^l$ | A sample of $\hat{\tau}_{i,j}$: $\mathbb{1}\{\tilde{x}_i^l > \mathbb{E}_n[\tilde{X}_i], \tilde{x}_j^l > \mathbb{E}_n[\tilde{X}_j]\}$ |
| **For Mixed Pair** $X_i, \tilde{X}_j$ | |
| $\tau_{i,j}$ | Probability of both $X_i$ and $\tilde{X}_j$ larger than their mean: $\mathbb{P}(X_i > 0, \tilde{X}_j > \mathbb{E}[\tilde{X}_j])$ |
| $\hat{\tau}_{i,j}$ | Estimation of $\tau_{i,j}$ : $\frac{1}{n}\sum_{l=1}^n \mathbb{1}\{x_i^l > 0, \tilde{x}_j^l > \mathbb{E}_n[\tilde{X}_j]\}$ |
| $\hat{\tau}_{i,j}^l$ | A sample of $\hat{\tau}_{i,j}$: $\mathbb{1}\{\tilde{x}_i^l > 0, \tilde{x}_j^l > \mathbb{E}_n[\tilde{X}_j]\}$ |

# B PROOF AND DERIVATIONS

## B.1 PROOF OF THM.2.1

If the $X_1, X_2$ and $X_3$ are jointly Gaussian and $X_1 \perp\!\!\!\perp X_3|X_2$, we have

$$Cov(X_1, X_3|X_2) = 0.$$

To test if $X_1, X_3$ are conditional independent given $\tilde{X}_2$, we are interested if $Cov(X_1, X_3|\tilde{X}_2)$ equals zero. Using the law of total covariance, we have

$$Cov(X_1, X_3|\tilde{X}_2) = \mathbb{E}[Cov(X_1, X_3|X_2, \tilde{X}_2)|\tilde{X}_2] + Cov(\mathbb{E}[X_1|X_2, \tilde{X}_2], \mathbb{E}[X_3|X_2, \tilde{X}_2]|\tilde{X}_2). \tag{12}$$

Since $\tilde{X}_2$ is the deterministic function of $X_2$, $\tilde{X}_2$ will be conditional independent with $X_1$ and $X_3$ given $X_2$. Therefore,

$$Cov(X_1, X_3|X_2, \tilde{X}_2) = Cov(X_1, X_3|X_2) = 0.$$

The first term of equation 12 is zero. We now focus on the second term. Similarly, we have

$$\mathbb{E}[X_1|X_2, \tilde{X}_2] = \mathbb{E}[X_1|X_2], \quad \mathbb{E}[X_3|X_2, \tilde{X}_2] = \mathbb{E}[X_3|X_2],$$

due to the conditional independence. One can see

$$Cov(X_1, X_3|X_2, \tilde{X}_2) = Cov(E[X_1|X_2], \mathbb{E}[X_3|X_2]|\tilde{X}_2).$$

Without loss of generality, we assume the mean of $X_1, X_2$ and $X_3$ are zero. Then $\mathbb{E}[X_1|X_2]$ and $\mathbb{E}[X_3|X_2]$ are scaled versions of $X_2$. The original equation becomes

$$Cov(X_1, X_3|X_2, \tilde{X}_2) = c \cdot Var(X_2|\tilde{X}_2),$$

where $c$ is a constant. We know that

$$Var(X_2|\tilde{X}_2) = \mathbb{E}[(X_2 - \mathbb{E}[X_2|\tilde{X}_2])^2|\tilde{X}_2],$$

which will be zero if and only if $X_2$ is almost surely a function of $\tilde{X}_2$. That means given $\tilde{X}_2$, the value of $X_2$ is determined exactly without any randomness, which clearly doesn't hold true in our discretization framework. Thus, $X_1 \not\perp\!\!\!\perp X_3|\tilde{X}_2$, which completes the proof.

## B.2 PROOF OF $\hat{\theta} \xrightarrow{p} \theta_0$

**Lemma B.1.** *For the estimation $\hat{\boldsymbol{\theta}} = (\hat{\sigma}_{i,j}, \hat{h}_i, \hat{h}_j), (\hat{\sigma}_{i,j}, \hat{h}_j), (\hat{\sigma}_{i,j})$ for discretized pairs, mixed pairs and continuous pairs respectively, calculated using bridge equation3 and equation6, will converge in probability to $\boldsymbol{\theta}_0 = (\sigma_{i,j}, h_i, h_j), (\sigma_{i,j}, h_j), (\sigma_{i,j})$ respectively.*

*Proof* According to the law of large numbers, for the estimated boundary $\hat{h}_i$ and $\hat{h}_j$ whose calculations are defined as $\hat{h}_j = \Phi^{-1}(1 - \hat{\tau}_j)$, we should have

$$n \to \infty, \quad \hat{\tau}_j = \frac{1}{n} \sum_{l=1}^{n} \mathbb{1}\{\tilde{x}_j^l > \mathbb{E}_n[\tilde{X}_j]\} \xrightarrow{p} \mathbb{P}(\tilde{X}_j > \mathbb{E}[\tilde{X}_j]).$$

Recall the definition $\mathbb{P}(\tilde{X}_j > E[\tilde{X}_j]) = 1 - \Phi(h_j)$, according to continuous mapping theorem (Vaart, 1998a), as long as the function $\Phi^{-1}(1 - \cdot)$ is continuous, we should have $\hat{h}_j \xrightarrow{p} h_j$. And thus $\hat{h}_i \xrightarrow{p} h_i, \hat{h}_j \xrightarrow{p} h_j$.

We further note that $\hat{\tau}_{i,j} = \bar{\Phi}(\hat{h}_i, \hat{h}_j, \hat{\sigma}_{i,j})$ and the estimation $\hat{\sigma}_{i,j}$ can be obtained through solving the function. Similarly, we also have

$$n \to \infty, \quad \hat{\tau}_{i,j} = \frac{1}{n} \sum_{l=1}^{n} \mathbb{1}\{\tilde{x}_i^l > \mathbb{E}_n[\tilde{X}_i]\}\mathbb{1}\{\tilde{x}_j^l > \mathbb{E}_n[\tilde{X}_j]\} \xrightarrow{p} \mathbb{P}(\tilde{x}_i^l > E[\tilde{X}_i], \tilde{x}_j^l > E[\tilde{X}_j])$$

$$= \tau_{i,j}.$$

According to the continuous mapping theorem, we have $\hat{\sigma}_{i,j} \xrightarrow{p} \sigma_{i,j}$. Thus, the parameter $(\hat{\sigma}_{i,j}, \hat{h}_i, \hat{h}_j) \xrightarrow{p} (\sigma_{i,j}, h_i, h_j)$ for the discretized pair case.

Apparently, the result above could easily extend to the mixed case where we fix $\hat{h}_i = h_i = 0$. Using the same procedure, we should have $(\hat{\sigma}_{i,j}, \hat{h}_j) \xrightarrow{p} (\sigma_{i,j}, h_j)$.

For the continuous case whose estimated variance is calculated as $\hat{\sigma}_{i,j} = \frac{1}{n}\sum_{l=1}^{n} x_i^l x_j^l - \frac{1}{n}\sum_{l=1}^{n} x_i^l \frac{1}{n}\sum_{l=1}^{n} x_j^l$, according to law of large numbers, we should have

$$n \to \infty, \quad \hat{\sigma}_{i,j} = \frac{1}{n}\sum_{l=1}^{n} x_i^l x_j^l - \frac{1}{n}\sum_{l=1}^{n} x_i^l \frac{1}{n}\sum_{l=1}^{n} x_j^l \xrightarrow{p} \mathbb{E}[X_i X_j] - \mathbb{E}[X_i]\mathbb{E}[X_j] = \sigma_{i,j}.$$

### B.3 PROOF OF ONE-TO-ONE MAPPING BETWEEN $\hat{\tau}_{i,j}$ AND $\hat{\sigma}_{i,j}$

**Lemma B.2.** *For any fixed $\hat{h}_i$ and $\hat{h}_j$, $T(\sigma'_{i,j}; \{\hat{h}_i, \hat{h}_j\}) = \int_{x_1 > \hat{h}_i} \int_{x_2 > \hat{h}_j} \phi(x_i, x_j; \sigma'_{i,j}) dx_i dx_j$, is a strictly monotonically increasing function on $\sigma'_{i,j} \in (-1, 1)$.*

*Proof* To prove the lemma, we just need to show the gradient $\frac{\partial T(\sigma'_{i,j}; \{\hat{h}_i, \hat{h}_j\})}{\partial \sigma} > 0$ for $\sigma'_{i,j} \in (-1, 1)$.

$$\frac{\partial T(\sigma_{i,j}); \{\hat{h}_i, \hat{h}_j\}}{\partial \sigma'_{i,j}} == \frac{1}{2\pi\sqrt{(1 - \sigma'^2_{i,j})}} \exp\left(-\frac{(\hat{h}_i^2 - 2\sigma'_{i,j}\hat{h}_i\hat{h}_j + \hat{h}_j^2)}{2(1 - \sigma'^2_{i,j})}\right),$$

which is obviously positive for $\sigma'_{i,j} \in (-1, 1)$. Thus, we have one-to-one mapping between $\hat{\tau}_{i,j}$ with the calculated $\hat{\sigma}_{i,j}$ for fixed $\hat{h}_i$ and $\hat{h}_j$.

### B.4 PROOF OF THM. 3.3

In this section, we provide the proof of Thm. 3.3, which utilizes a regular statistical tool: Z-estimator (Vaart, 1998b). Specifically, we are interested in the parameter $\boldsymbol{\theta}$ and we have it estimation $\hat{\boldsymbol{\theta}}$. Let $\boldsymbol{x_1}, \ldots, \boldsymbol{x_n}$ are sampled from some distribution, we can construct the function characterized by the parameter $\boldsymbol{\theta}$ related the $\boldsymbol{x}$ as $\boldsymbol{\psi_\theta}(\boldsymbol{x})$. As long as we have $n$ observations, we can construct the function as follows

$$\Psi_n(\boldsymbol{\theta}) = \frac{1}{n}\sum_{l=1}^{n} \boldsymbol{\psi_\theta}(\boldsymbol{x}_i) = \mathbb{E}_n[\boldsymbol{\psi_\theta}].$$

We further specify the form

$$\Psi(\boldsymbol{\theta}) = \int \boldsymbol{\psi_\theta}(\boldsymbol{x}) d\boldsymbol{x} = \mathbb{E}[\boldsymbol{\psi_\theta}].$$

**Assume the estimator $\hat{\theta}$ is a zero of $\Psi_n$, i.e., $\Psi_n(\hat{\theta}) = 0$ and will converge in probability to $\boldsymbol{\theta}_0$, which is a zero of $\Psi$, i.e., $\Psi(\boldsymbol{\theta}_0) = 0$.** Expand $\Psi_n(\hat{\theta})$ in a Taylor series around $\boldsymbol{\theta}_0$, we should have

$$0 = \Psi_n(\hat{\boldsymbol{\theta}}) = \Psi_n(\boldsymbol{\theta}_0) + (\hat{\boldsymbol{\theta}} - \boldsymbol{\theta}_0)\Psi'_n(\boldsymbol{\theta}_0) + \frac{1}{2}(\hat{\boldsymbol{\theta}} - \boldsymbol{\theta}_0)^2 \Psi''_n(\boldsymbol{\theta}_0) + \ldots$$

Rearrange the equation above, we have

$$\hat{\boldsymbol{\theta}} - \boldsymbol{\theta}_0 = -\frac{\Psi_n(\boldsymbol{\theta}_0)}{\Psi'_n(\boldsymbol{\theta}_0) + \frac{1}{2}(\hat{\boldsymbol{\theta}} - \boldsymbol{\theta}_0)^2 \Psi''_n(\boldsymbol{\theta}_0)}$$

$$= -\frac{\frac{1}{n}\sum_{l=1}^{n} \boldsymbol{\psi_\theta}(\boldsymbol{x}_i)}{\Psi'_n(\boldsymbol{\theta}_0) + \frac{1}{2}(\hat{\boldsymbol{\theta}} - \boldsymbol{\theta}_0)^2 \Psi''_n(\boldsymbol{\theta}_0)}.$$

According to the central limit theorem, the numerator will be asymptotic normal with variance $\mathbb{E}[\psi_{\boldsymbol{\theta}_0}^2]/n$ as the mean $\Psi(\boldsymbol{\theta}_0) = 0$ is zero. The first term of denominator $\Psi'_n(\boldsymbol{\theta}_0)$ will converge in probability to $\Psi'(\boldsymbol{\theta}_0)$ according to the law of large numbers. The second term $\hat{\boldsymbol{\theta}} - \boldsymbol{\theta}_0 = o_P(1)$. [2] As long as the denominator converges in probability and the numerator converges in distribution, according to Slusky's lemma, we have

$$\sqrt{n}(\hat{\boldsymbol{\theta}} - \boldsymbol{\theta}_0) \xrightarrow{d} N\left(0, \frac{\mathbb{E}[\psi_{\boldsymbol{\theta}_0}^2]}{\mathbb{E}[\psi'_{\boldsymbol{\theta}_0}]^2}\right). \tag{13}$$

Extend into the high-dimensional case we should have

$$\hat{\boldsymbol{\theta}} - \boldsymbol{\theta}_0 = -\Psi'_n(\boldsymbol{\theta}_0)^{-1}\Psi_n(\boldsymbol{\theta}_0)$$

where the second order term is omitted, further assume the matrix $\mathbb{E}[\psi'_{\boldsymbol{\theta}_0}]$ is invertible, we have

$$\sqrt{n}(\hat{\boldsymbol{\theta}} - \boldsymbol{\theta}_0) \xrightarrow{d} N\left(0, (\mathbb{E}[\psi'_{\boldsymbol{\theta}_0}])^{-1}\mathbb{E}[\psi_{\boldsymbol{\theta}_0}\psi_{\boldsymbol{\theta}_0}^T](\mathbb{E}[\psi'^T_{\boldsymbol{\theta}_0}])^{-1}\right), \tag{14}$$

Specifically, in our case $\boldsymbol{\theta}_0 = (\sigma_{i,j}, \boldsymbol{\Lambda})$, where $\boldsymbol{\Lambda}$ is another parameter set influencing the estimation of $\sigma_{i,j}$ (will discuss case in case in later proof). In the practical scenario, we only have access to the estimated parameter $\hat{\boldsymbol{\theta}}$ and the empirical distribution $\mathbb{P}_n$, which will converge to their true counterparts. Thus, we have

$$\hat{\sigma}_{i,j} - \sigma_{i,j} \overset{\text{approx}}{\sim} N\left(0, ((\mathbb{E}_n[\psi'_{\hat{\boldsymbol{\theta}}}])^{-1}\mathbb{E}_n[\psi_{\hat{\boldsymbol{\theta}}}\psi_{\hat{\boldsymbol{\theta}}}^T](\mathbb{E}_n[\psi'^T_{\hat{\boldsymbol{\theta}}}])^{-1})_{1,1}\right).$$

Under the null hypothesis of independent, $\sigma_{i,j} = 0$. We provide the proof that $\hat{\boldsymbol{\theta}} \xrightarrow{p} \boldsymbol{\theta}_0$ of our case in App. B.2. Thus, $\mathbb{E}_n[\psi_{\hat{\boldsymbol{\theta}}}]$, the function parameterized by $\hat{\boldsymbol{\theta}}$, should also converge in $\mathbb{E}_n[\psi_{\boldsymbol{\theta}_0}]$ when $n \to \infty$. Besides, by the law of large numbers, $\mathbb{E}_n[\psi_{\hat{\boldsymbol{\theta}}_0}]$ will converge to $\mathbb{E}[\psi_{\hat{\boldsymbol{\theta}}_0}]$. Thus, the equation above will converge to equation 14 when $n \to \infty$.

We note that the construction of Z-estimator above require two necessary ingredients: 1. The estimated parameter $\hat{\boldsymbol{\theta}}$ should be the zero of the sample mean of criterion function $\Psi_n$. 2. The estimated parameter $\hat{\boldsymbol{\theta}}$ should converge in probability to $\boldsymbol{\theta}_0$, the zero of the true mean of criterion function $\Psi$. For different cases (discretized, mixed, continuous), the construction of criterion function varies. We provide their corresponding derivation in Lem. 3.5, 3.6, 3.4 respectively.

### B.5 Derivation of Lem. 3.5

Let's first focus on the most challenging case where both variables are discretized observations and our interested parameter will include $\hat{\boldsymbol{\theta}} = (\hat{\sigma}_{i,j}, \hat{h}_i, \hat{h}_j)$ (Although we only care about the distribution of $\hat{\sigma}_{i,j} - \sigma_{i,j}$, the estimation of boundary $\hat{h}_i$ and $\hat{h}_j$ will influence the estimation of $\hat{\sigma}_{i,j}$, thus we need to consider all of them).

The next step will be to *construct an appropriate criterion function $\boldsymbol{\psi}$ such that $\Psi_n(\hat{\boldsymbol{\theta}}) = \mathbf{0}$*. Given $n$ observations $\{\tilde{\boldsymbol{x}}^1, \tilde{\boldsymbol{x}}^2, \ldots, \tilde{\boldsymbol{x}}^n\}$, which are discretized version of $\{\boldsymbol{x}^1, \boldsymbol{x}^2, \ldots, \boldsymbol{x}^n\}$ we should have

$$\Psi_n(\hat{\boldsymbol{\theta}}) = \begin{pmatrix} \Psi_n(\hat{\sigma}_{i,j}) \\ \Psi_n(\hat{h}_i) \\ \Psi_n(\hat{h}_j) \end{pmatrix} = \frac{1}{n}\sum_{l=1}^n \boldsymbol{\psi}_{\hat{\boldsymbol{\theta}}}(\tilde{\boldsymbol{x}}^l) = \frac{1}{n}\sum_{l=1}^n \begin{pmatrix} \hat{\tau}_{i,j}^l - T(\hat{\sigma}_{i,j}; \{\hat{h}_i, \hat{h}_j\}) \\ \hat{\tau}_i^l - \bar{\Phi}(\hat{h}_i) \\ \hat{\tau}_j^l - \bar{\Phi}(\hat{h}_j) \end{pmatrix} = \mathbf{0}. \tag{15}$$

$$\Psi_n(\boldsymbol{\theta}_0) = \begin{pmatrix} \Psi_n(\sigma_{i,j}) \\ \Psi_n(h_i) \\ \Psi_n(h_j) \end{pmatrix} = \frac{1}{n}\sum_{l=1}^n \boldsymbol{\psi}_{\boldsymbol{\theta}_0}(\tilde{\boldsymbol{x}}^l) = \frac{1}{n}\sum_{l=1}^n \begin{pmatrix} \hat{\tau}_{i,j}^l - T(\sigma_{i,j}; \{h_i, h_j\}) \\ \hat{\tau}_i^l - \bar{\Phi}(h_i) \\ \hat{\tau}_j^l - \bar{\Phi}(h_j) \end{pmatrix}. \tag{16}$$

---

[2]We will not provide proof of this in this paper; however, interested readers may refer to (Vaart, 1998b)

The difference between the estimated parameter with the true parameter can be expressed as

$$\hat{\boldsymbol{\theta}} - \boldsymbol{\theta}_0 = \begin{pmatrix} \hat{\sigma}_{i,j} - \sigma_{i,j} \\ \hat{h}_i - h_i \\ \hat{h}_j - h_j \end{pmatrix} = -\frac{1}{n} \sum_{l=1}^{n} \begin{pmatrix} \frac{\partial \Psi_n(\sigma_{i,j})}{\partial \sigma_{i,j}} & \frac{\partial \Psi_n(\sigma_{i,j})}{\partial h_i} & \frac{\partial \Psi_n(\sigma_{i,j})}{\partial h_j} \\ \frac{\partial \Psi_n(h_i)}{\partial \sigma_{i,j}} & \frac{\partial \Psi_n(h_i)}{\partial h_i} & \frac{\partial \Psi_n(h_i)}{\partial h_j} \\ \frac{\partial \Psi_n(h_j)}{\partial \sigma_{i,j}} & \frac{\partial \Psi_n(h_j)}{\partial h_i} & \frac{\partial \Psi_n(h_j)}{\partial h_j} \end{pmatrix}^{-1}$$
$$\cdot \begin{pmatrix} \hat{\tau}_{i,j}^l - T(\sigma_{i,j}; \{h_i, h_j\}) \\ \hat{\tau}_i^l - \bar{\Phi}(h_i) \\ \hat{\tau}_j^l - \bar{\Phi}(h_j) \end{pmatrix}, \tag{17}$$

where the specific form of each entry of the gradient matrix is expressed as

$$\frac{\partial \Psi_n(\sigma_{i,j})}{\partial \sigma_{i,j}} = -\frac{1}{2\pi\sqrt{(1-\sigma_{i,j}^2)}} \exp\left(-\frac{(h_i^2 - 2\sigma_{i,j}h_ih_j + h_j^2)}{2(1-\sigma_{i,j}^2)}\right);$$

$$\frac{\partial \Psi_n(\sigma_{i,j})}{\partial h_i} = \int_{h_j}^{\infty} \frac{1}{2\pi\sqrt{1-\sigma_{i,j}^2}} \exp\left(-\frac{h_i^2 - 2\sigma_{i,j}h_ix_2 + x_2^2}{2(1-\sigma_{i,j}^2)}\right) dx_2;$$

$$\frac{\partial \Psi_n(\sigma_{i,j})}{\partial h_j} = \int_{h_i}^{\infty} \frac{1}{2\pi\sqrt{1-\sigma_{i,j}^2}} \exp\left(-\frac{h_j^2 - 2\sigma_{i,j}h_jx_1 + x_1^2}{2(1-\sigma_{i,j}^2)}\right) dx_1;$$

$$\frac{\partial \Psi_n(h_i)}{\partial \sigma_{i,j}} = 0;$$

$$\frac{\partial \Psi_n(h_i)}{\partial h_i} = \frac{1}{\sqrt{2\pi}} \exp\left(-\frac{h_i^2}{2}\right); \tag{18}$$

$$\frac{\partial \Psi_n(h_i)}{\partial h_j} = 0;$$

$$\frac{\partial \Psi_n(h_j)}{\partial \sigma_{i,j}} = 0;$$

$$\frac{\partial \Psi_n(h_j)}{\partial h_i} = 0;$$

$$\frac{\partial \Psi_n(h_j)}{\partial h_j} = \frac{1}{\sqrt{2\pi}} \exp\left(-\frac{h_j^2}{2}\right).$$

For simplicity of notation, we define

$$\hat{\sigma}_{i,j} - \sigma_{i,j} = \frac{1}{n} \sum_{l=1}^{n} \xi_{i,j}^l,$$

where the specific form is of $\{\xi_{i,j}^l\}$ is defined in equation 17. We should note that $\{\xi_{i,j}^l\}$ are i.i.d random variables with mean zero (this property will be the key to the derivation of inference of CI). As long as our estimation $\hat{\boldsymbol{\theta}}$ converge in probability to $\boldsymbol{\theta}_0$ as proved in B.2, we have

$$\sqrt{n}(\hat{\boldsymbol{\theta}} - \boldsymbol{\theta}_0) \xrightarrow{d} N\left(0, ((\mathbb{E}[\boldsymbol{\psi}_{\boldsymbol{\theta}_0}'])^{-1}\mathbb{E}[\boldsymbol{\psi}_{\boldsymbol{\theta}_0}\boldsymbol{\psi}_{\boldsymbol{\theta}_0}^T](\mathbb{E}[\boldsymbol{\psi}_{\boldsymbol{\theta}_0}'^T])^{-1})\right),$$

where $\boldsymbol{\psi}_{\boldsymbol{\theta}_0}$ is defined in equation 16. However, in practice, we don't have access to either $\boldsymbol{\theta}_0$ or the true expectation. In this scenario, we can plug in the sample mean of $\mathbb{E}_n[\boldsymbol{\psi}_{\hat{\boldsymbol{\theta}}}]$ to get the estimated variance, i.e., the actual variance used in the calculation of $\hat{\sigma}_{i,j} - \sigma_{i,j}$ is

$$\frac{1}{n}\left((\mathbb{E}_n[\boldsymbol{\psi}_{\hat{\boldsymbol{\theta}}}'])^{-1}\mathbb{E}_n[\boldsymbol{\psi}_{\hat{\boldsymbol{\theta}}}\boldsymbol{\psi}_{\hat{\boldsymbol{\theta}}}^T](\mathbb{E}_n[\boldsymbol{\psi}_{\hat{\boldsymbol{\theta}}}'^T])^{-1}\right)_{1,1}. \tag{19}$$

### B.6 DERIVATION OF LEM. 3.6

Use the same procedure as in the derivation of Lem. 3.5, for mixed pair of observations where $X_i$ is continuous and $\tilde{X}_j$ is discrete, we can construct the criterion function

$$\Psi_n(\hat{\boldsymbol{\theta}}) = \begin{pmatrix} \Psi_n(\hat{\sigma}_{i,j}) \\ \Psi_n(\hat{h}_j) \end{pmatrix} = \frac{1}{n}\sum_{l=1}^n \boldsymbol{\psi}_{\hat{\boldsymbol{\theta}}}(\tilde{\boldsymbol{x}}^l) = \frac{1}{n}\sum_{l=1}^n \begin{pmatrix} \hat{\tau}_{i,j}^l - T(\hat{\sigma}_{i,j}; \{0, \hat{h}_j\}) \\ \hat{\tau}_j^l - \bar{\Phi}(\hat{h}_j) \end{pmatrix} = \mathbf{0}. \tag{20}$$

$$\Psi_n(\boldsymbol{\theta}_0) = \begin{pmatrix} \Psi_n(\sigma_{i,j}) \\ \Psi_n(h_j) \end{pmatrix} = \frac{1}{n}\sum_{l=1}^n \boldsymbol{\psi}_{\boldsymbol{\theta}_0}(\tilde{\boldsymbol{x}}^l) = \frac{1}{n}\sum_{l=1}^n \begin{pmatrix} \hat{\tau}_{i,j}^l - T(\sigma_{i,j}; \{0, h_j\}) \\ \hat{\tau}_j^l - \bar{\Phi}(h_j) \end{pmatrix}. \tag{21}$$

The difference between the estimated parameter with the true parameter can be expressed as

$$\hat{\boldsymbol{\theta}} - \boldsymbol{\theta}_0 = \begin{pmatrix} \hat{\sigma}_{i,j} - \sigma_{i,j} \\ \hat{h}_j - h_j \end{pmatrix} = -\frac{1}{n}\sum_{l=1}^n \begin{pmatrix} \frac{\partial \Psi_n(\sigma_{i,j})}{\partial \sigma_{i,j}} & \frac{\partial \Psi_n(\sigma_{i,j})}{\partial h_j} \\ \frac{\partial \Psi_n(h_j)}{\partial \sigma_{i,j}} & \frac{\partial \Psi_n(h_j)}{\partial h_j} \end{pmatrix}^{-1} \begin{pmatrix} \hat{\tau}_{i,j}^l - T(\sigma_{i,j}; \{0, h_j\}) \\ \hat{\tau}_j^l - \bar{\Phi}(h_j). \end{pmatrix}, \tag{22}$$

where the specific form of each entry of the gradient matrix can be found in equation 18. Using exactly the same procedure, we should have the same formation of the variance calculated as equation 19 with a different definition of $\boldsymbol{\psi}_{\boldsymbol{\theta}_0}$ and $\boldsymbol{\psi}_{\hat{\boldsymbol{\theta}}}$ defined in equation 21 equation 20.

### B.7 DERIVATION OF LEM. 3.4

Use the same line of procedure as in the derivation of Lem. 3.5, for a continuous pair of variables, we can construct the criterion function

$$\Psi_n(\hat{\boldsymbol{\theta}}) = \Psi_n(\hat{\sigma}_{i,j}) = \frac{1}{n}\sum_{l=1}^n x_i^l x_j^l - \frac{1}{n}\sum_{l=1}^n x_i^l \frac{1}{n}\sum_{l=1}^n x_j^l - \hat{\sigma}_{i,j} = 0. \tag{23}$$

$$\Psi_n(\boldsymbol{\theta}_0) = \Psi_n(\sigma_{i,j}) = \frac{1}{n}\sum_{l=1}^n x_i^l x_j^l - \frac{1}{n}\sum_{l=1}^n x_i^l \frac{1}{n}\sum_{l=1}^n x_j^l - \sigma_{i,j}.$$

Denote $\frac{1}{n}\sum_{l=1}^n x_i^l$ as $\bar{x}_i$ and $\frac{1}{n}\sum_{l=1}^n x_j^l$ as $\bar{x}_j$. We should have

$$\hat{\sigma}_{i,j} - \sigma_{i,j} = \frac{1}{n}\sum_{l=1}^n x_i^l x_j^l - \bar{x}_i \bar{x}_j - \sigma_{i,j}. \tag{24}$$

According to equation 13, we have

$$\sqrt{n}(\hat{\sigma}_{i,j} - \sigma_{i,j}) \rightsquigarrow N\left(0, \frac{\mathbb{E}[\boldsymbol{\psi}_{\boldsymbol{\theta}_0}^2]}{(\mathbb{E}[\boldsymbol{\psi}_{\boldsymbol{\theta}_0}'])^2}\right).$$

where $(E[\boldsymbol{\psi}_{\boldsymbol{\theta}_0}'])^2 = 1$. In practical calculation, we have the variance

$$\frac{1}{n}\mathbb{E}_n[\boldsymbol{\psi}_{\hat{\boldsymbol{\theta}}}^2]/(\mathbb{E}_n[\boldsymbol{\psi}_{\hat{\boldsymbol{\theta}}}'])^2 = \frac{1}{n^2}\sum_{l=1}^n (x_i^l x_j^l - \bar{x}_i \bar{x}_j - \hat{\sigma}_{i,j})^2.$$

### B.8 PROOF OF THM. 3.8

#### B.8.1 PROOF OF LEM. 3.7

Consider our latent continuous variables $\boldsymbol{X} = (X_1, \ldots, X_p) \sim N(0, \boldsymbol{\Sigma})$ and do nodewise regression

$$X_j = \boldsymbol{X}_{-j}\boldsymbol{\beta}_j + \epsilon_j,$$

where $\boldsymbol{X}_{-j}$ is the submatrix of $\boldsymbol{X}$ with $X_j$ removed. We can divide its covariance $\boldsymbol{\Sigma}$ and its precision matrix $\Omega = \boldsymbol{\Sigma}^{-1}$ into the predictor $\boldsymbol{X}_{-j}$ and outcome variable $X_j$ in our regression:

$$\boldsymbol{\Sigma} = \begin{pmatrix} \boldsymbol{\Sigma}_{j,j} & \boldsymbol{\Sigma}_{j,-j} \\ \boldsymbol{\Sigma}_{-j,j} & \boldsymbol{\Sigma}_{-j,-j} \end{pmatrix} \quad \boldsymbol{\Omega} = \begin{pmatrix} \boldsymbol{\Omega}_{j,j} & \boldsymbol{\Omega}_{j,-j} \\ \boldsymbol{\Omega}_{-j,j} & \boldsymbol{\Omega}_{-j,-j} \end{pmatrix} .$$

Just like regular linear regression, we can get

$$n \to \infty, \quad \boldsymbol{\beta}_j = \boldsymbol{\Sigma}_{-j,-j}^{-1} \boldsymbol{\Sigma}_{-j,j}.$$

From the invertibility of a block matrix

$$\begin{bmatrix} A & B \\ C & D \end{bmatrix}^{-1} = \begin{bmatrix} (A - BD^{-1}C)^{-1} & -(A - BD^{-1}C)^{-1}BD^{-1} \\ -D^{-1}C(A - BD^{-1}C)^{-1} & D^{-1} + D^{-1}C(A - BD^{-1}C)^{-1}BD^{-1} \end{bmatrix}.$$

If $A$ and $D$ is invertible, we will have

$$\begin{bmatrix} A & B \\ C & D \end{bmatrix}^{-1} = \begin{bmatrix} (A - BD^{-1}C)^{-1} & 0 \\ 0 & (D - CA^{-1}B)^{-1} \end{bmatrix} \begin{bmatrix} I & -BD^{-1} \\ -CA^{-1} & I \end{bmatrix}.$$

Thus, we can get:

$$\boldsymbol{\Omega}_{j,j} = (\boldsymbol{\Sigma}_{j,j} - \boldsymbol{\Sigma}_{j,-j}\boldsymbol{\Sigma}_{-j,-j}^{-1}\boldsymbol{\Sigma}_{-j,j})^{-1};$$

$$\boldsymbol{\Omega}_{j,-j} = -(\boldsymbol{\Sigma}_{j,j} - \boldsymbol{\Sigma}_{j,-j}\boldsymbol{\Sigma}_{-j,-j}^{-1}\boldsymbol{\Sigma}_{-j,j})^{-1}\boldsymbol{\Sigma}_{j,-j}(\boldsymbol{\Sigma}_{-j,-j})^{-1}.$$

Move one step forward:

$$-\boldsymbol{\Omega}_{j,j}^{-1}\boldsymbol{\Omega}_{j,-j} = \boldsymbol{\Sigma}_{j,-j}(\boldsymbol{\Sigma}_{-j,-j})^{-1}.$$

Take transpose for both sides, as long as $\boldsymbol{\Omega}$ is a symmetric matrix and $\boldsymbol{\Omega}_{-j,j} = \boldsymbol{\Omega}_{j,-j}^T$, we will have

$$-\boldsymbol{\Omega}_{j,j}^{-1}\boldsymbol{\Omega}_{-j,j} = \boldsymbol{\Sigma}_{-j,-j}^{-1}\boldsymbol{\Sigma}_{-j,j} = \boldsymbol{\beta}_j.$$

We should note testing $\boldsymbol{\Omega}_{-j,j} = 0$ is equivalent to testing $\boldsymbol{\beta}_j = 0$ as the $\boldsymbol{\Omega}_{j,j}$ will always be nonzero. The variable $\boldsymbol{\Omega}_{-j,j}$ captures the CI of $X_j$ with other variables. As long as the variable $\boldsymbol{\Omega}_{j,j}$ is just one scalar, we can get

$$\beta_{j,k} = -\frac{\omega_{j,k}}{\omega_{j,j}}$$

capturing the CI relationship between variable $X_j$ with $X_k$ conditioning on all other variables.

### B.8.2 DETAILED DERIVATION OF INFERENCE FOR $\boldsymbol{\beta_j}$

Nodewise regression allows us to use the regression parameter $\boldsymbol{\beta}_j$ as the surrogate of $\Omega_{-j,j}$. The problem now transfers to constructing the inference for $\boldsymbol{\beta}_j$, specifically, the derivation of distribution of $\hat{\boldsymbol{\beta}}_j - \boldsymbol{\beta}_j$. The overarching concept is that we are already aware of the distribution of $\hat{\sigma}_{i,j} - \sigma_{i,j}$ and we know that there exists a deterministic relationship between $\boldsymbol{\beta}_j$ with $\boldsymbol{\Sigma}$. Consequently, we can express $\hat{\boldsymbol{\beta}}_j - \boldsymbol{\beta}_j$ as a composite of $\hat{\sigma}_{i,j} - \sigma_{i,j}$ to establish such an inference. Specifically, we have

$$\begin{aligned} \hat{\boldsymbol{\beta}}_j - \boldsymbol{\beta}_j &= \hat{\boldsymbol{\Sigma}}_{-j,-j}^{-1}\hat{\boldsymbol{\Sigma}}_{-j,j} - \boldsymbol{\Sigma}_{-j,-j}^{-1}\boldsymbol{\Sigma}_{-j,j} \\ &= \hat{\boldsymbol{\Sigma}}_{-j,-j}^{-1}\left(\hat{\boldsymbol{\Sigma}}_{-j,j} - \hat{\boldsymbol{\Sigma}}_{-j,-j}\boldsymbol{\Sigma}_{-j,-j}^{-1}\boldsymbol{\Sigma}_{-j,j}\right) \\ &= -\hat{\boldsymbol{\Sigma}}_{-j,-j}^{-1}\left(\hat{\boldsymbol{\Sigma}}_{-j,-j}\boldsymbol{\beta}_j - \boldsymbol{\Sigma}_{-j,-j}\boldsymbol{\beta}_j + \boldsymbol{\Sigma}_{-j,-j}\boldsymbol{\beta}_j - \hat{\boldsymbol{\Sigma}}_{-j,j}\right) \\ &= -\hat{\boldsymbol{\Sigma}}_{-j,-j}^{-1}\left((\hat{\boldsymbol{\Sigma}}_{-j,-j} - \boldsymbol{\Sigma}_{-j,-j})\boldsymbol{\beta}_j - (\hat{\boldsymbol{\Sigma}}_{-j,j} - \boldsymbol{\Sigma}_{-j,j})\right), \end{aligned}$$

where each entry in matrix $(\hat{\boldsymbol{\Sigma}}_{-j,-j} - \boldsymbol{\Sigma}_{-j,-j})$ and $(\hat{\boldsymbol{\Sigma}}_{-j,j} - \boldsymbol{\Sigma}_{-j,j})$ denotes the difference between estimated covariance with true covariance.

For ease of notation, we further denote that

$$\hat{\sigma}_{i,j} - \sigma_{i,j} = \frac{1}{n}\sum_{l=1}^{n}\xi_{i,j}^l,$$

where $\xi^l_{i,j}$ are i.i.d random variables with specific form defined in equation 17 for discrete case, equation 22 for mixed case and equation 24 in continuous case.

Suppose that we want to test the CI of the variable $X_1$ with other variables, $j = 1$. We then have

$$
\hat{\boldsymbol{\Sigma}}_{-1,-1} - \boldsymbol{\Sigma}_{-1,-1} = \begin{bmatrix} \hat{\sigma}_{2,2} \ldots \hat{\sigma}_{2,p} \\ \ldots \\ \hat{\sigma}_{p,2} \ldots \hat{\sigma}_{p,p} \end{bmatrix} - \begin{bmatrix} \sigma_{2,2} \ldots \sigma_{2,p} \\ \ldots \\ \sigma_{p,2} \ldots \sigma_{p,p} \end{bmatrix} = \frac{1}{n} \sum_{l=1}^{n} \begin{bmatrix} \xi^l_{2,2} \ldots \xi^l_{2,p} \\ \ldots \\ \xi^l_{p,2} \ldots \xi^l_{p,p} \end{bmatrix},
$$

$$
\hat{\boldsymbol{\Sigma}}_{-1,1} - \boldsymbol{\Sigma}_{-1,1} = \begin{bmatrix} \hat{\sigma}_{2,1} \\ \ldots \\ \hat{\sigma}_{p,1} \end{bmatrix} - \begin{bmatrix} \sigma_{2,1} \\ \ldots \\ \sigma_{p,1} \end{bmatrix} = \frac{1}{n} \sum_{l=1}^{n} \begin{bmatrix} \xi^l_{2,1} \\ \ldots \\ \xi^l_{p,1} \end{bmatrix}.
$$

Put them together:

$$
\hat{\boldsymbol{\beta}}_1 - \boldsymbol{\beta}_1 = \begin{bmatrix} \hat{\beta}_{1,2} - \beta_{1,2} \\ \hat{\beta}_{1,3} - \beta_{1,3} \\ \ldots \\ \hat{\beta}_{1,p} - \beta_{1,p} \end{bmatrix} = -\hat{\boldsymbol{\Sigma}}^{-1}_{-1,-1} \frac{1}{n} \sum_{l=1}^{n} \left( \begin{bmatrix} \xi^l_{2,2} & \xi^l_{2,3} & \cdots & \xi^l_{2,p} \\ \xi^l_{3,2} & \xi^l_{3,3} & \cdots & \xi^l_{3,p} \\ \ldots & \ldots & \ldots & \ldots \\ \xi^l_{p,2} & \xi^l_{p,3} & \cdots & \xi^l_{p,p} \end{bmatrix} \begin{bmatrix} \beta_{1,2} \\ \beta_{1,3} \\ \ldots \\ \beta_{1,p} \end{bmatrix} - \begin{bmatrix} \xi^l_{2,1} \\ \xi^l_{3,1} \\ \ldots \\ \xi^l_{p,1} \end{bmatrix} \right).
$$

As $\frac{1}{n} \sum_{l=1}^{n} \xi^l_{i,j}$ is asymptotically normal, the who vector of $\hat{\boldsymbol{\beta}}_1 - \boldsymbol{\beta}_1$ is a linear combination of Gaussian distribution. However, We cannot merely engage in a linear combination of its variance as they are dependent with each other. For example, if $Y_1, Y_2$ are dependent and we are trying to find out $Var(aY_1 + bY_2)$, we should have

$$
Var(aY_1 + bY_2) = \begin{bmatrix} a & b \end{bmatrix} \begin{bmatrix} Var(Y_1) & Cov(Y_1, Y_2) \\ Cov(Y_1, Y_2) & Var(Y_2) \end{bmatrix} \begin{bmatrix} a \\ b \end{bmatrix}. \tag{25}
$$

Now, suppose we are interested in the distribution of $\hat{\beta}_{1,2} - \beta_{1,2}$, we have

$$
\hat{\beta}_{1,2} - \beta_{1,2} = -\frac{1}{n} \sum_{l=1}^{n} (\hat{\boldsymbol{\Sigma}}^{-1}_{-1,-1})_{[2],:} \left( \begin{bmatrix} \xi^l_{2,2} & \xi^l_{2,3} & \cdots & \xi^l_{2,p} \\ \xi^l_{3,2} & \xi^l_{3,3} & \cdots & \xi^l_{3,p} \\ \ldots & \ldots & \ldots & \ldots \\ \xi^l_{p,2} & \xi^l_{p,3} & \cdots & \xi^l_{p,p} \end{bmatrix} \begin{bmatrix} \beta_{1,2} \\ \beta_{1,3} \\ \ldots \\ \beta_{1,p} \end{bmatrix} - \begin{bmatrix} \xi^l_{2,1} \\ \xi^l_{3,1} \\ \ldots \\ \xi^l_{p,1} \end{bmatrix} \right),
$$

where $(\hat{\boldsymbol{\Sigma}}^{-1}_{-1,-1})_{[2],:}$ is the row of index of $X_2$ of $\hat{\boldsymbol{\Sigma}}^{-1}_{-1,-1}$ ([2] denotes the index of the variable, e.g., $(\hat{\boldsymbol{\Sigma}}^{-1}_{-1,-1})_{[2],:}$ represents the first row of $\hat{\boldsymbol{\Sigma}}^{-1}_{-1,-1}$ since the row of first variable is removed. ). For ease of notation, we define

$$
\boldsymbol{Y}^l := \boldsymbol{\Xi}^l_{-1,-1} = \begin{bmatrix} \xi^l_{2,2} & \xi^l_{2,3} & \cdots & \xi^l_{2,p} \\ \xi^l_{3,2} & \xi^l_{3,3} & \cdots & \xi^l_{3,p} \\ \ldots & \ldots & \ldots & \ldots \\ \xi^l_{p,2} & \xi^l_{p,3} & \cdots & \xi^l_{p,p} \end{bmatrix} \in \mathbb{R}^{p-1 \times p-1}, \qquad \boldsymbol{v}^l := \boldsymbol{\Xi}^l_{-1,1} = \begin{bmatrix} \xi^l_{2,1} \\ \xi^l_{3,1} \\ \ldots \\ \xi^l_{p,1} \end{bmatrix} \in \mathbb{R}^{p-1},
$$

and

$$
\boldsymbol{u} := (\hat{\boldsymbol{\Sigma}}^{-1}_{-1,-1})^T_{[2],:} \in \mathbb{R}^{p-1} \qquad \boldsymbol{w} := \begin{bmatrix} \beta_{1,2} \\ \beta_{1,3} \\ \ldots \\ \beta_{1,p} \end{bmatrix} \in \mathbb{R}^{p-1}.
$$

We can rewrite the equation as

$$
\hat{\beta}_{1,2} - \beta_{1,2} = -\frac{1}{n} \sum_{l=1}^{n} \boldsymbol{u}(\boldsymbol{Y}^l \boldsymbol{w} - \boldsymbol{v}^l).
$$

We note that $\boldsymbol{Y}^l$, $\boldsymbol{v}^l$ are variables, and $\boldsymbol{u}$, $\boldsymbol{w}$ are constants (just like the example $aY_1 + bY_2$). We further let $m = p - 1$ to simplify the notation. We can thus write the equation above as vector form:

$$\hat{\beta}_{1,2} - \beta_{1,2} = -\frac{1}{n}\sum_{l=1}^{n}[u_1, \ldots, u_m, u_1w_1, u_1w_2, \ldots, u_mw_m]\begin{bmatrix} -v_1^l \\ \ldots, \\ -v_m^l \\ Y_{11}^l \\ Y_{12}^l \\ \ldots \\ Y_{mm}^l \end{bmatrix}$$

$$= -\frac{1}{n}\sum_{l=1}^{n}[\boldsymbol{u}^T, \text{vec}(\boldsymbol{u}\boldsymbol{w}^T)^T]\begin{bmatrix} -\boldsymbol{v}^l \\ \text{vec}(\boldsymbol{Y})^l \end{bmatrix},$$

where $u_i$ represents the $i$-th element of vector $\boldsymbol{u}$ and $Y_{ij}^l$ represents the entry in $i$-th row and $j$-th column of matrix $\boldsymbol{Y}^l$, vec represents the row-wise vectorization of a matrix, e.g,

$$\text{vec}(\boldsymbol{Y}^l) = \begin{bmatrix} Y_{11} \\ Y_{12} \\ Y_{13} \\ \ldots \\ Y_{mm} \end{bmatrix} \in \mathbb{R}^{m^2}.$$

Similar as equation 25, the variance is calculated as

$$Var\left(\sqrt{n}(\hat{\beta}_{1,2} - \beta_{1,2})\right) = \frac{1}{n}\sum_{l=1}^{n}[\boldsymbol{u}^T, \text{vec}(\boldsymbol{u}\boldsymbol{w}^T)^T]\begin{bmatrix} -\boldsymbol{v}^l \\ \text{vec}(\boldsymbol{Y})^l \end{bmatrix}\begin{bmatrix} -\boldsymbol{v}^l \\ \text{vec}(\boldsymbol{Y})^l \end{bmatrix}^T\begin{bmatrix} \boldsymbol{u} \\ \text{vec}(\boldsymbol{u}\boldsymbol{w}^T) \end{bmatrix}.$$

Now we go back to use the notations of $\xi$ and $\boldsymbol{\Sigma}$. Under the null hypothesis that $X_1 \perp\!\!\!\perp X_2|X_{others}$, i.e., $\beta_{1,2} = 0$. We thus use $\tilde{\boldsymbol{\beta}}_1$ to denote $\boldsymbol{\beta}_1$ where $\beta_{1,2} = 0$. Let

$$B_{-1}^l = \begin{pmatrix} \xi_{2,1}^l & \xi_{3,1}^l & \cdots & \xi_{p,1}^l \\ \xi_{2,2}^l & \xi_{2,3}^l & \cdots & \xi_{2,p}^l \\ \xi_{3,2}^l & \xi_{3,3}^l & \cdots & \xi_{3,p}^l \\ \cdots & \cdots & \cdots & \cdots \\ \xi_{p,2}^l & \xi_{p,3}^l & \cdots & \xi_{p,p}^l \end{pmatrix} = \begin{bmatrix} \boldsymbol{\Xi}_{-1,1}^l{}^T \\ \boldsymbol{\Xi}_{-1,-1}^l \end{bmatrix},$$

and

$$\boldsymbol{a}^{[2]} = \begin{bmatrix} -(\hat{\boldsymbol{\Sigma}}_{-1,-1}^{-1})_{[2],:}^T \\ \text{vec}\left((\hat{\boldsymbol{\Sigma}}_{-1,-1}^{-1})_{[2],:}^T\tilde{\boldsymbol{\beta}}_1^T\right) \end{bmatrix}$$

Similarly as equation 25, The variance is calculated as

$$Var\left(\sqrt{n}(\hat{\beta}_{1,2} - \beta_{1,2})\right) = \boldsymbol{a}^{[2]T}\frac{1}{n}\sum_{l=1}^{n}\text{vec}(\boldsymbol{B}_{-1}^l)\text{vec}(\boldsymbol{B}_{-1}^l)^T\boldsymbol{a}^{[2]},$$

Simply replace the index $1, 2$ as general index $j, k$, the distribution of $\hat{\beta}_{j,k} - \beta_{j,k}$ is

$$\hat{\beta}_{j,k} - \beta_{j,k} \xrightarrow{d} N(0, \boldsymbol{a}^{[k]T}\frac{1}{n^2}\sum_{l=1}^{n}\text{vec}(\boldsymbol{B}_{-j}^l)\text{vec}(\boldsymbol{B}_{-j}^l)^T)\boldsymbol{a}^{[k]}).$$

In practice, we can plug in the estimates of $\beta_j$ to estimate the interested distribution and do the CI test by hypothesizing $\beta_{j,k} = 0$.

### B.9 DISCUSSION OF ASSUMPTION

In this section, we first justify why the assumption of zero mean and identity variance can be made without loss of generality. Then, we explain the rationale behind the linear Gaussian assumption.

#### B.9.1 ZERO MEAN AND IDENTITY VARIANCE

In this section, we engage in a more thorough discussion regarding our assumptions about $\boldsymbol{X}$. Specifically, we demonstrate that this assumption of mean and variance does not compromise the generality. In other words, the true model may possess different mean and variance values, but we proceed by treating it as having a mean of zero and identity variance.

The key ingredient allowing us to assume such a model is, the discretization function $g_j$ is an unknown nonlinear monotonic function. Suppose the $g_j'$ maps the continuous domain to a binary variable, and we have the "groundtruth" variable, denoted $X_j'$, with mean $a$ and variance $b$. Assume the cardinality of the discretized domain is only 2, i.e., our observation $\tilde{X}_j$ can only be 0 or 1. We further have the constant $d_j'$ as the discretization boundary such that we have the observation

$$\tilde{X}_j = \mathbb{1}(g_j'(X_j') > d_j') = \mathbb{1}(X_j' > g_j'^{-1}(d_j)).$$

We can always produce our assumed variable $X_j$ with mean 0 and variance 1, such that $X_j = \frac{1}{\sqrt{b}}X_j' - \frac{a}{\sqrt{b}}$ and the same observation with a different nonlinear transformation $g_j$ and decision boundary $d_j$, such that

$$\tilde{X}_j = \mathbb{1}(g_j(X_j) > d_j) = \mathbf{1}(X_j > g_j^{-1}(d_j)) = \mathbb{1}(X_j' > \sqrt{b}g_j^{-1}(d_j) + a).$$

As long as the observation $\tilde{X}_j$ is the same, we should have $\sqrt{b}g_j^{-1}(d_j) + a = g_j'^{-1}(d_j)$. Our assumed model $X_j$ clearly mimics the "groundtruth" $X_j'$. Besides, according to Lem. B.3, we have one-to-one mapping between $\hat{\tau}_{i,j}$ with the estimated covariance for fixed $\hat{h}_i, \hat{h}_j$. Thus, as long as the observation is the same, the estimation of covariance $\hat{\sigma}_{i,j}$ remains unaffected by our assumptions regarding the mean and variance of $\boldsymbol{X}$, so do the following inference.

We further conduct casual discovery experiments to empirically validate our statement, which is shown in App. H.3.

#### B.9.2 DISCUSSION OF LINEAR GAUSSIAN ASSUMPTION

Discretization of continuous variables inevitably leads to information loss in the original data. Compared to the original distributional information, the recovered covariance matrix is naturally less accurate. Given this, constructing a valid statistical inference procedure, rather than solely relying on estimated covariance values for drawing conditional independence conclusions, is desirable.

One major limitation of DCT is its reliance on the assumption that latent continuous variables follow a multivariate normal distribution. Violations of this assumption can lead to erroneous conclusions. For instance, consider a scenario where the relationship between latent variables is nonlinear, such as $X_i = X_j^2$. In this case, the covariance $\sigma_{i,j}$ equals zero despite a deterministic dependency between $X_i$ and $X_j$. Consequently, even if the correlation is perfectly estimated, the model fails to capture the true underlying relationship, leading to incorrect inferences.

Nevertheless, although the theoretical framework of DCT requires latent continuous variables to follow a multivariate Gaussian distribution, experimental results in various settings, even in situations in which this assumption is violated, demonstrate the usefulness and robustness of DCT, suggesting the development of this technique is essential to causal discovery from discretized continuous data. Further details of the empirical validations are provided in Appendix H.

## C FIGURE OF MAIN EXPERIMENTS: CAUSAL DISCOVERY

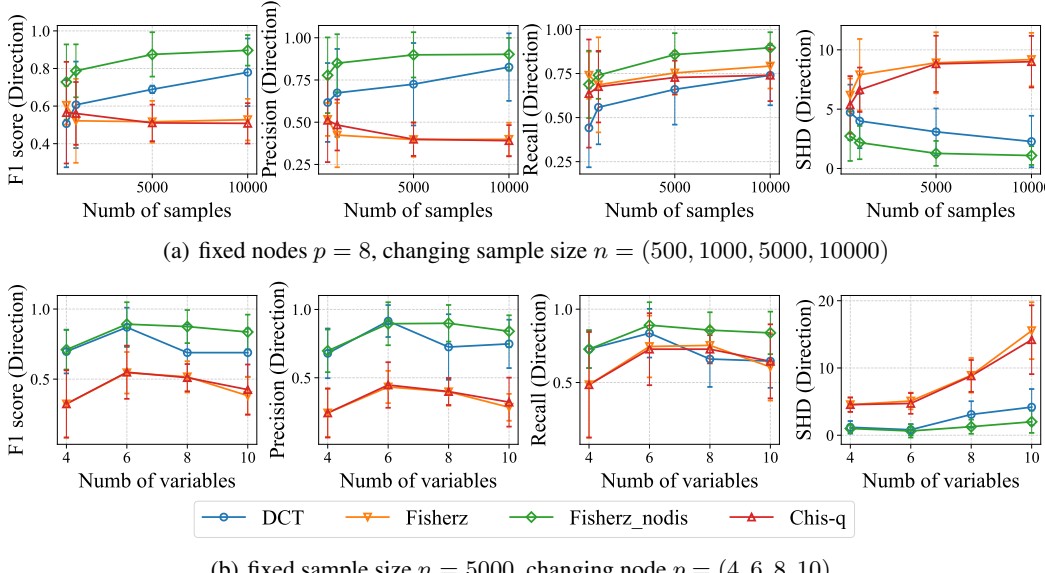

(a) fixed nodes $p = 8$, changing sample size $n = (500, 1000, 5000, 10000)$

(b) fixed sample size $n = 5000$, changing node $p = (4, 6, 8, 10)$

Figure 4: Experiment result of DAG discovery on synthetic data for changing sample size (a) and changing number of nodes (b). Fisherz_nodis is the Fisher-z test applied to original continuous data. We evaluate $F_1$ (↑), Precision (↑), Recall (↑) and SHD (↓).

# D  PSEUDO CODE

---

**Algorithm 1** DCT: Discretization-Aware CI Test

---

1: **Require:**
- Observed data matrix $\tilde{\boldsymbol{X}}' \in \mathbb{R}^{n \times d}$
- Tested pair indices $i, j$ with $i \neq j$
- Conditioning set $\mathbf{S} \subseteq \{1, \ldots, d\} \setminus \{i, j\}$
- Significance level $\alpha$

2: **Rearrange Data Matrix**

$$\tilde{\boldsymbol{X}} = \left[ \tilde{\boldsymbol{X}}'[:,i], \tilde{\boldsymbol{X}}'[:,j], \tilde{\boldsymbol{X}}'[:,\mathbf{S}] \right] \in \mathbb{R}^{n \times p}, \quad \text{where } p = 2 + |\mathbf{S}|$$

3: **Initialize Covariance Matrix**

$$\hat{\boldsymbol{\Sigma}} \leftarrow \mathbf{I}_p \quad \text{(identity matrix of size } p \times p\text{)}$$

4: **for** $q \leftarrow 1$ **to** $p$ **do**
5:     **for** $k \leftarrow q + 1$ **to** $p$ **do**
6:         **if** both $\tilde{X}[:,q]$ and $\tilde{X}[:,k]$ are continuous **then**
7:             Compute sample covariance $\hat{\sigma}_{q,k}$ using equation 5
8:         **else**
9:             Compute covariance $\hat{\sigma}_{q,k}$ using Equation equation 6
10:         **end if**
11:         Update covariance matrix:

$$\hat{\boldsymbol{\Sigma}}[q,k] \leftarrow \hat{\sigma}_{q,k}$$

$$\hat{\boldsymbol{\Sigma}}[k,q] \leftarrow \hat{\sigma}_{q,k} \quad \text{(ensuring symmetry)}$$

12:     **end for**
13: **end for**
14: **Extract Submatrices ($i$ and $j$ correspond the first and second column of $\tilde{X}$ due to the regroup)**
- Let $\hat{\boldsymbol{\Sigma}}_{-1,-1} \in \mathbb{R}^{p-1 \times p-1} \leftarrow$ the submatrix of $\hat{\boldsymbol{\Sigma}}$ without 1st column and 1st row
- Let $\hat{\boldsymbol{\Sigma}}_{-1,1} \in \mathbb{R}^{p-1}$ be the 1st column of $\hat{\boldsymbol{\Sigma}}$ with first row removed

15: **Compute Test Statistics**

$$\hat{\beta}_{1,2} \leftarrow \hat{\boldsymbol{\Sigma}}_{-1,-1}^{-1} \hat{\boldsymbol{\Sigma}}_{-1,1}$$

16: **Formulate Null Distribution**

$$\Phi(z) \leftarrow \text{Cumulative distribution function of the Normal Distribution defined in Thm. 3.8}$$

17: **Calculate P-value**

$$p\text{-value} \leftarrow 2 \cdot \left( 1 - \Phi\left( |\hat{\beta}_{1,2}| \right) \right)$$

18: **Make Decision**
19: **if** $p$-value $> \alpha$ **then**
20:     **Conclude**: $X_i \perp\!\!\!\perp X_j \mid X_{\mathbf{S}}$
21: **else**
22:     **Conclude**: $X_i \not\perp\!\!\!\perp X_j \mid X_{\mathbf{S}}$
23: **end if**
24: **return** The conditional independence decision

---

# E  RELATED WORK

Testing for CI is pivotal in the field of causal discovery (Spirtes et al., 2000), and a variety of methods exist for performing CI tests (CI tests). An important group of CI test methods involves the assumption of Gaussian variables with linear dependencies. For example, under this assumption, Gaussian graphical models are extensively studied (Yuan and Lin, 2007; Peterson et al., 2015; Mohan et al., 2012; Ren et al., 2015). To address CI test under Gaussian assumption, partial correlation serves as a viable method for CI testing (Baba et al., 2004). To evaluate the independence of variables $X_1$ and $X_2$ conditional on $\boldsymbol{Z}$, The technique proposed by (Su and White, 2008) determines CI by comparing the estimations of $p(X_1|X_2, \boldsymbol{Z})$ and $p(X_1|X_2)$.

Another approach involves discretizing $\boldsymbol{Z}$ and performing independent tests within each resulting bin (Margaritis, 2005). Our work, however, diverges from these existing methods in two significant ways. Firstly, we are equipped to handle data, where partial variables are discretized. Additionally, we postulate that discrete variables are derived from the transformation of continuous variables in a latent Gaussian model. With the same assumption, the most closely related study is by (Fan et al., 2017), where the authors developed a novel rank-based estimator for the precision matrix of mixed data. However, their work stops short of providing a CI test for this method. Our research fills this gap, offering the ability to estimate the precision matrix for both discrete and mixed data and providing a rigorous CI test for our methodology.

Recent advancements in CI testing have utilized kernel methods for continuous variables influenced by nonlinear relationships. (Fukumizu et al., 2004) describes non-parametric CI relationships using covariance operators in reproducing kernel Hilbert spaces (RKHS). KCI test (Zhang et al., 2012) assesses the partial associations of regression functions linking $x$, $y$, and $z$, while RCI test (Strobl et al., 2019) aims to enhance the KCI test's efficiency. In KCIP test (Doran et al., 2014) employs permutations of samples to emulate CI scenarios. CCI test (Sen et al., 2017) further reformulates testing into a process that leverages the capabilities of supervised learning models. For discrete variable analysis, the $G^2$ test (Aliferis et al., 2010) and conditional mutual information (Zhang et al., 2010) are commonly employed. However, their method cannot deal with our setting where only discretized version of latent variables can be observed.

# F  RESOURCE USAGE

All the experiments are run using Intel(R) Xeon(R) CPU E5-2680 v4 with 55 processors. It costs 4 hours to run experiments in Section 3.1.

# G  LIMIATION AND BROADER IMPACTS

**Limitation**   So far, the largest limitation of our method is to treat discretized variables as binary, which wastes the available information. Besides that, the parametric assumption limits its generalizability. However, we need to point out this is pretty normal in CI test fields.

**Broader Impacts**   The goal of our proposed method is to test the conditional independence relationship given discretized observation. This task is essential and has broad applications. We are confident that our method will be beneficial and will not result in negative societal impacts.

# H ADDITIONAL EXPERIMENTS

## H.1 LINEAR NON-GAUSSIAN AND NONLINEAR

Our model requires that the original data must adhere to the hypothesis of following a multivariate normal distribution, which appears to potentially limit the generalizability. Therefore, it is worthwhile to explore its robustness when such assumptions are violated. In this regard, we conducted several experiments, including scenarios involving linear non-Gaussian and nonlinear Gaussian.

For both cases, we follow the setting of our experiment where there are $p = 8$ nodes and $p - 1$ edges. We explore the effect of changing sample size $n = (100, 500, 2000, 5000)$. Specifically for linear non-Gaussian case, we adhere to some of the settings outlined by (Shimizu et al., 2011), conducting experiments where the original continuous data followed: (1) a Student's t-distribution with 3 degrees of freedom, (2) a uniform distribution, and (3) an exponential distribution. Each variable is generated as $X_i = f(PA_i) + noise$, where $noise$ follows the distribution in (1), (2), (3) correspondingly and $f$ is an arbitrary linear function. The first three rows of Fig. 5 and Fig. 6 show the result of the linear non-Gaussian case.

For the nonlinear cases, we follow setting in (Li et al., 2024), where every variable $X_i$ is generated as $X_i = f(WPA_i + noise)$, $noise \sim N(0, 1)$ and $f$ is a function randomly chosen from (a) $f(x) = sin(x)$, (b) $f(x) = x^3$, (c) $f(x) = tanh(x)$, and (d) $f(x) = ReLU(x)$. $W$ is a linear function. Similarly, we set the number of nodes at $p = 8$ and change the number of samples $n = (500, 2000, 5000)$. For both cases, we run 10 graph instances with different seeds and report the result of skeleton discovery in Fig. 5 and DAG in Fig. 6 (The same orientation rules (Dor and Tarsi, 1992) used in the main experiment are employed to convert a CPDAG (Chandler Squires, 2018) into a DAG). The last row of Fig. 5 and Fig. 6 shows the result of the nonlinear case.

Based on the experimental outcomes, DCT demonstrates marginally superior or comparable efficacy in terms of the F1-score, precision, and SHD relative to both the Fisher-Z test and the Chi-square test when dealing with small sample sizes. Nevertheless, as the sample size increases, DCT's performance clearly surpasses that of the aforementioned tests across all three evaluated metrics, especially in the linear case. Consistent with observations from the main experiment, DCT exhibits a lower recall in comparison to the baseline tests. This discrepancy can be attributed to the baseline tests being prone to incorrectly infer conditional dependence and connect a large proportion of nodes. According to the results, our test shows notable robustness under the case assumptions are violated, confirming its practical effectiveness.

## H.2 DENSER GRAPH

DCT primarily works on cases where CI is mistakenly judged as conditional dependence due to discretization. Consequently, its efficacy is more pronounced in scenarios characterized by a relatively sparse graph, as numerous instances are truly conditionally independent. Nevertheless, the investigation of causal discovery with a dense latent graph is essential for evaluating the power of a test, i.e., its ability to successfully reject the null hypothesis when the tested pairs are conditionally dependent. Thus, we conduct the experiment where $p = 8, n = 10000$ and changing edges $(p + 2, p + 4, p + 6)$. Similarly, the latent continuous data follows a multivariate Gaussian model and the true DAG $\mathcal{G}$ is constructed using BP model. We run 10 graph instances with different seeds and report the result of the skeleton discovery and DAG in Fig. 7.

According to the experiment results, DCT exhibits better performance in terms of the F1-score, precision, and SHD relative to both the Fisher-Z test and the Chi-square test. As the graph becomes progressively denser, the superiority of the DCT correspondingly diminishes as there are few conditional independent cases in the true DAG. Due to the same reason, The recall remains lower than that of other baseline methods.

## H.3 MULTIVARIATE GAUSSIAN WITH NONZERO MEAN AND NON-UNIT VARIANCE

We employed a setting nearly identical to the main experiment, with the only difference being the alteration in data generation: instead of using a standard normal distribution, we used a Gaussian distribution with mean sampled from $U(-2, 2)$ and variance sampled from $U(0, 3)$. We fix the number of variables as $p = 8$ and change the number of samples $n = (100, 500, 2000, 5000)$. The Fig. 8 shows the result and demonstrates the effectiveness of our method.

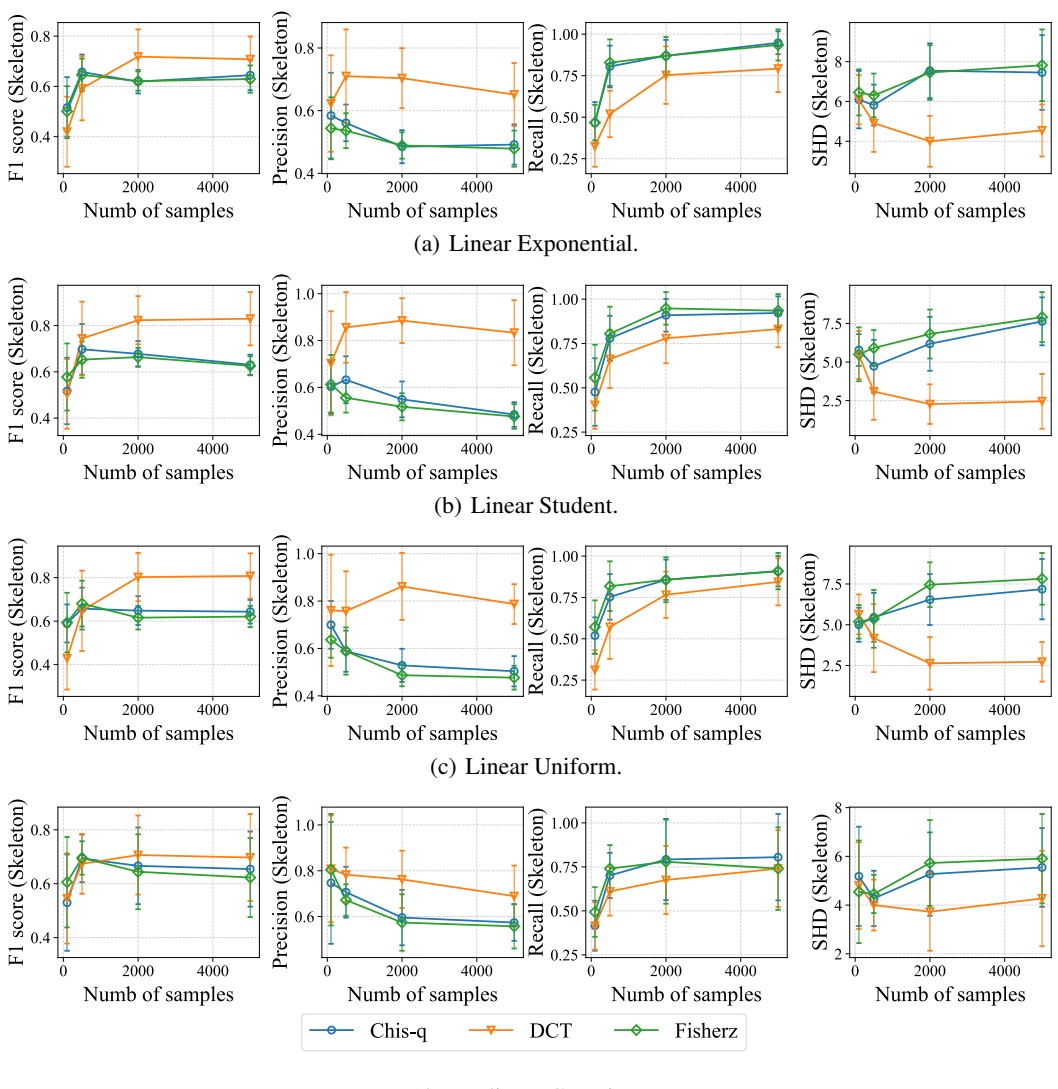

(a) Linear Exponential.

(b) Linear Student.

(c) Linear Uniform.

(d) Nonlinear Gaussian.

Figure 5: Experiment result of causal discovery on synthetic data with $p = 8$, $n = (100, 500, 2000, 5000)$ where the data generation process violates our assumptions. The data are generated with either nongaussian distributed (a), (b), (c) or the relations are not linear (d). The figure reports $F_1$ ($\uparrow$), Precision ($\uparrow$), Recall ($\uparrow$) and SHD ($\downarrow$) on skeleton.

## H.4   REAL-WORLD DATASET

To further validate DCT, we employ it on a real-world dataset: Big Five Personality https://openpsychometrics.org/, which includes 50 personality indicators and over 19000 data samples. Each variable contains 5 possible discrete values to represent the scale of the corresponding questions, where 1=Disagree, 2=Weakly disagree, 3=Neutral, 4=Weakly agree and 5=Agree, e.g., "N3=1" means "I agree that I worry about things". This scenario clearly suits DCT, where the degree of agreement with a certain question must be a continuous variable while we can only observe the result after categorization. We choose three variables respectively: [N3: I worry about things], [N10: I often feel blue ], [N4: I seldom feel blue]. We then do the casual discovery using PC algorithm with DCT and compare it with the Chi-square test and Fisher-Z test. The result can be found in Fig. 9.

Based on the experimental outcomes, despite the absence of a groundtruth for reference, we observe that the results obtained via DCT appear more plausible than those derived from Fisher-Z and Chi-square tests. Specifically, DCT suggests the relationship $N_3 \perp\!\!\!\perp N4|N10$, which is reasonable as intuitively, the answer of 'I often feel blue' already captures the information of 'I seldom feel blue'.

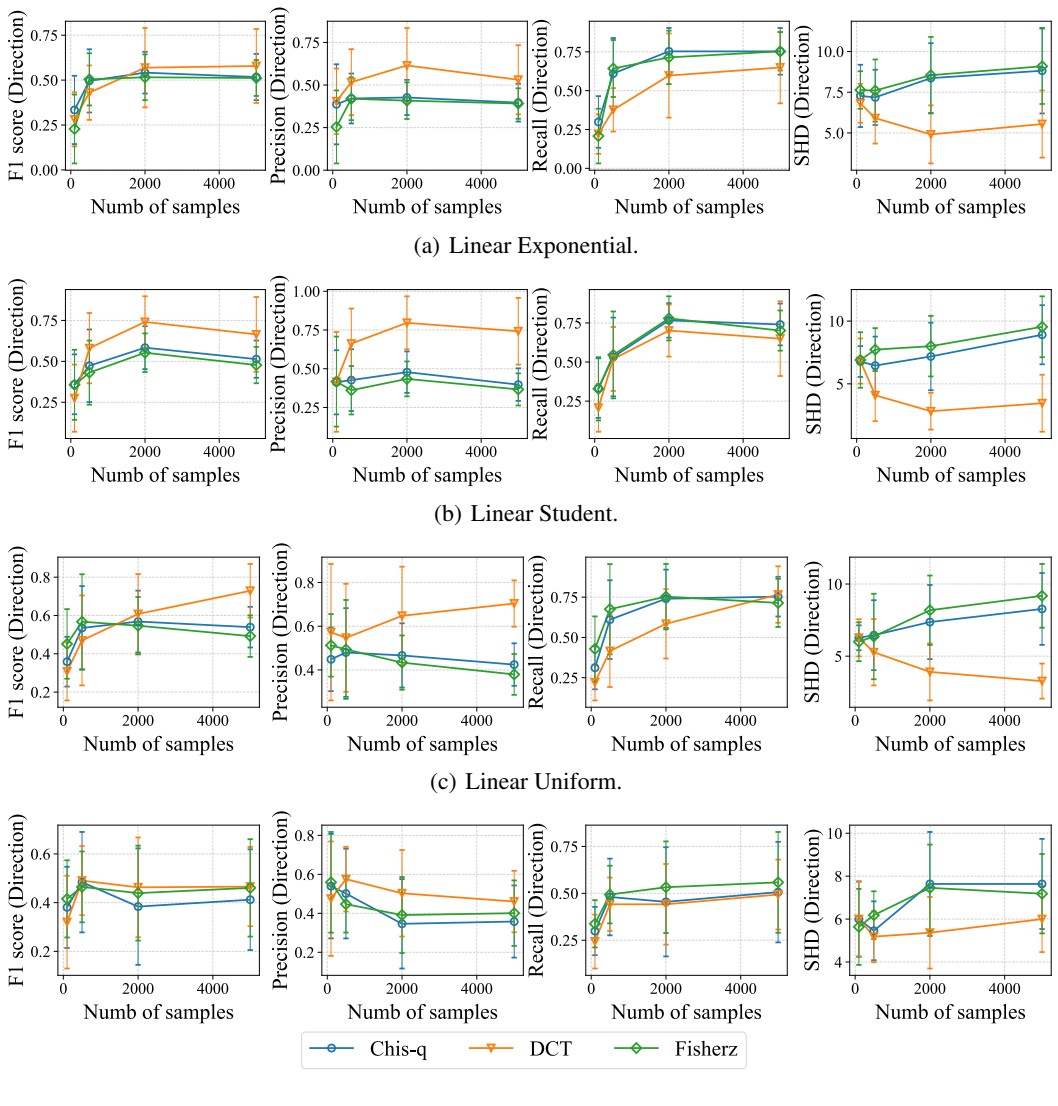

(a) Linear Exponential.

(b) Linear Student.

(c) Linear Uniform.

(d) Nonlinear Gaussian.

Figure 6: Experiment result of causal discovery on synthetic data with $p = 8$, $n = (100, 500, 2000, 5000)$ where the data generation process violates our assumptions. The data are generated with either nongaussian distributed (a), (b), (c) or the relations are not linear (d). The figure reports $F_1$ (↑), Precision (↑), Recall (↑) and SHD (↓) on DAG.

As a comparison, both Fisher-Z and Chi-square return a fully connected graph. The results directly correspond to our illustrative example shown in Fig. 1, substantiating the necessity of our proposed test.

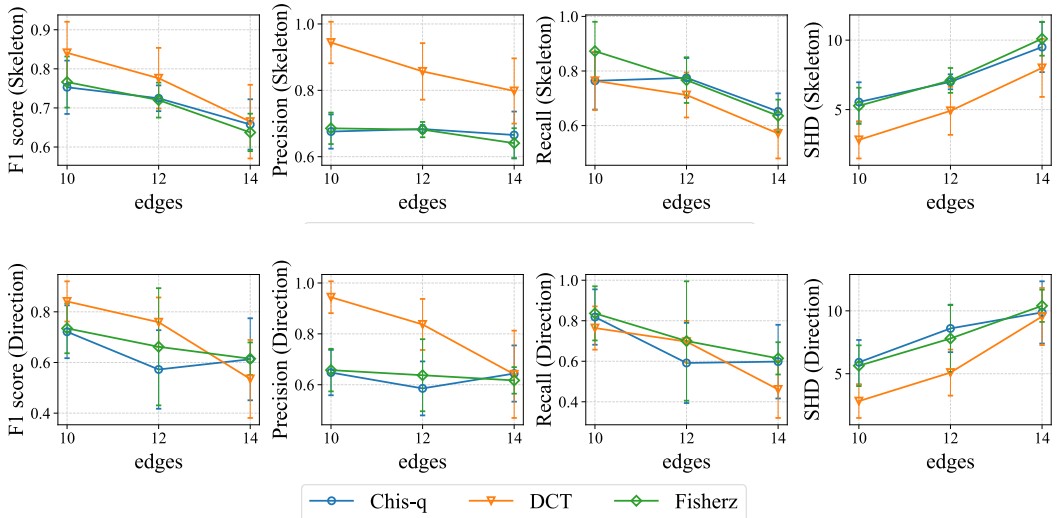

Figure 7: Experimental comparison of causal discovery on synthetic datasets for denser graphs with $p = 8, n = 10000$ and edges varying $p + 2, p + 4, p + 6$. We evaluate $F_1$ ($\uparrow$), Precision ($\uparrow$), Recall ($\uparrow$) and SHD ($\downarrow$) on both skeleton and DAG.

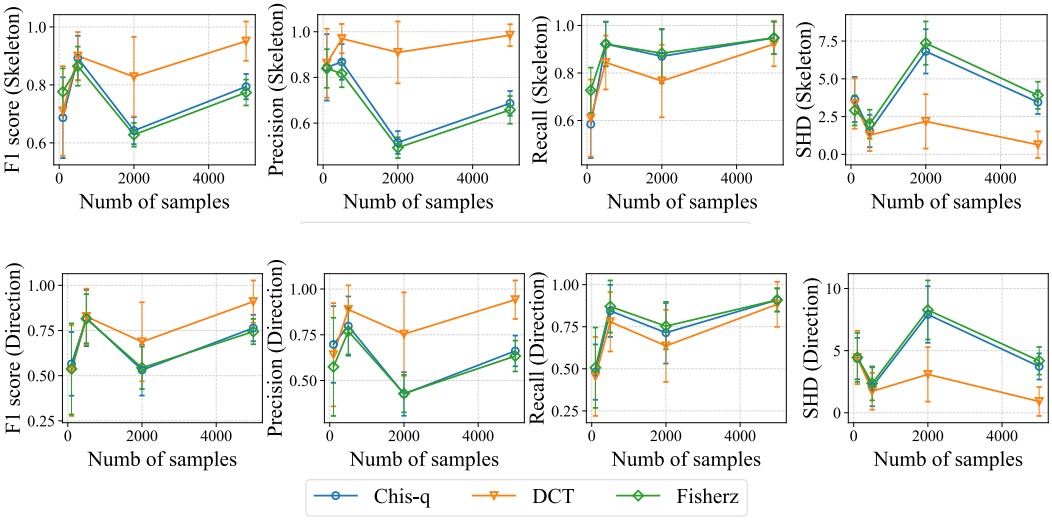

Figure 8: Experimental comparison of causal discovery on synthetic datasets for multivariate Gaussian model with $p = 8, n = (100, 500, 2000, 5000)$ and where mean is not zero. We evaluate $F_1$ ($\uparrow$), Precision ($\uparrow$), Recall ($\uparrow$) and SHD ($\downarrow$) on both skeleton and DAG.

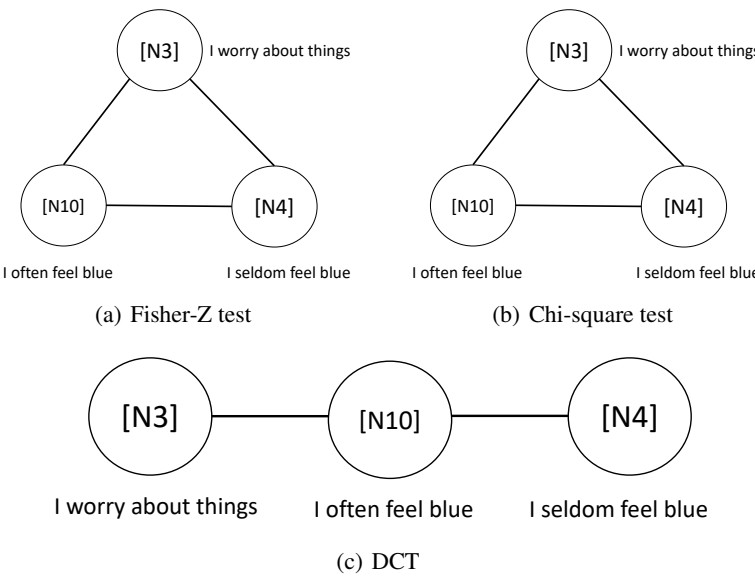

(a) Fisher-Z test

(b) Chi-square test

(c) DCT

Figure 9: Experimental comparison of causal discovery on the real-world dataset.

