# OpenReview forum: "A Conditional Independence Test in the Presence of Discretization"
_ICLR.cc/2025/Conference — ICLR 2025 Poster_

### Official Review · Reviewer_i1w6 · 2024-10-17

**Soundness:** 3
**Presentation:** 2
**Contribution:** 2
**Rating:** 3
**Confidence:** 3

**Summary:**

The paper addresses errors in existing CI tests caused by an implicit discretization of the original continuous variable during measurements. The authors propose DCT to deal with this issue and establish the necessary theoretical foundations under assumptions such as having a Gaussian distribution. The method is tested using simulations.

**Strengths:**

The studied problem is important. With respect to the presentation, the problem is well motivated, formulated, and explained, and the experimental result is also written clearly. Some good effort is put into the experiments.

**Weaknesses:**

1) The main drawback of this paper is the overall presentation. I was not able to grasp the main idea, preventing me from properly evaluating this work. The following provides some details:

a) The authors were careful and patient in explaining detailed notations up to line 190. However, after that, the pace and notational complexity suddenly got of hand. For example, in line 192, the notation 1{ } is not even defined, let alone the meaning of the equation (the sentence in line 194 does not help). It was as if this part of the paper was written by a different author. I had a hard time understanding the paper from this point.

b) Definition 3.1. What is “the estimated boundary”? It should be rigorously defined somewhere. If the intention of this definition is to define “the estimated boundary” then the language is not appropriate. It should be something like this: “the \emph{estimated boundary of variables $X_j$ is defined as ….”. Currently, it reads as the estimated boundary is already defined before.

c) Definition 3.4. Isn’t this simply the formula for the sample covariance? that is Cov(X,Y) = 1/n \sum_{i=1}^n (X_iY_i) - \mu_X\mu_Y? If so, what’s the point of giving it a separate definition and name (bridge equation)?

d) Theorem 3.5. I am confused by what is the rigorous statement in this theorem? By “this statistic is approximated to follow a normal distribution”, do the authors mean that the statistic converges to a normal distribution as N grows?

e) Theorem 3.9. Where is the \hat{\beta}_{jk} formally defined? I suppose the answer is not eq 13 as then it should have been out of the theorem.

f) The Experiment Section is back on track and written clearly. It was unclear though why the addition of W at line 450 is good for assessing the power.

2) One of the main motivations of this work was its application in Bayesian network (BN) structure learning. Nevertheless, insufficient effort was put into it. As the authors stated, the experiments in Section 4.2 are only for [extremely] sparse DAGs. They do consider denser ones in C2, but that is an only minor improvement, as the number of edges increased from p-1 to at most p+6. True DAGs can be much denser. The authors should provide a fair comparison for this part (also not to move these important parts in the appendix).

a) I suggest to test several established datasets in BN structure learning such as ASIA, CHILD, WATER, and ANDES.

b) Similar to the other experiment section, here, non-Gaussian distributions should be tested for structure learning.

c) Even if the results here are weak, DCT appears to be a promising method that is worthy of publication, provided a major rewriting.

d) Note that without this structure learning application, this paper perhaps better suits a statistics journal.

---
Overall, a good problem is studied in this paper with a seemingly effective solution. However, the presentation of the technical parts needs a substantial improvement.

**Questions:**

Minor comments:

line 73) This is a very interesting idea to use a DAG to invest the discretization problem; however, the authors implicitly assume that the DAG is faithful. The fact that $X_1$ is not d-separated of $X_3$ given $\tilde X_2$in the DAG does not imply that the $X_1\not\perp X_3\mid \tilde X_2$ in the distribution.
line 73) extra comma
line 89) 1). 2). 3). are not common. Consider 1) 2) 3)
line 100) So all discretized variables are Gaussian? This is a limitation.
[important] line 137) The notation seems to be redundantly complex. Why \sigma_{j_1,j_2} rather than just \sigma_{j,k}, or even simpler \sigma_{jk}?
line 159) better not to start a sentence with an equation.
line 264) Consider using ^\top for transpose.
- Was DCT ever defined? What does it stand for?
line 264) grammar “contain”
line 269) “null hypothesis”
line 270) Be more specific, what are “our settings” and “our framework”?

---

> ### Author Response · Authors · 2024-11-25
>
> We sincerely thank the reviewer for the **detailed and thorough** review, which has helped us realize that the organization of our original submission lacked clarity and may have caused confusion. In response, we have revised our paper with the following key improvements:
>
> 1. **Notation Table**: We have simplified the notation to avoid unnecessary complexity. Additionally, we have included a complete table of notations in Appendix A and provided definitions for the key notations at the beginning of Section 3 to enhance readability and accessibility.
>
> 2. **Algorithm Pseudocode**: To enhance clarity, we now include the pseudocode of our core algorithm, DCT, in Appendix~D, offering a clearer understanding of its workflow and implementation.
>
> 3. **Reorganization and Intuition**: We have reorganized the sections on the **bridge equation**, **calculation of covariance**, and **independence tests**. In these sections, we have introduced new notations where necessary and made an effort to provide intuitive explanations alongside the formal mathematical definitions.
> 4. We formulate the preliminary of nodewise regression as a lemma to highlight the conclusion it provides direct link to the regression parameter $\beta$ with the covariance $\mathbf{\Sigma}$ and precision matrix $\mathbf{\Omega}$.
>
>
> We hope these changes address the reviewer’s concerns and significantly improve the paper’s clarity and structure.

---

> > ### Author Response · Authors · 2024-11-25
> > **Point-to-Point rebuttal**
> >
> > Q.  the notation 1{ } is not even defined.
> >
> > A: We are sorry for the confusion. $1\{condition\}$ is the indicator function which returns 1 if condition is held, 0 otherwise. The definition is provided in line 132-133 and Appendix~A in our revised version for easier reference.
> >
> > > Q.  What is “the estimated boundary”?
> >
> > A: We apologize for the lack of clarity. Estimated boundary is $\hat{h}_j$ which corresponds to the estimation of $h_j$, which is defined as the value in the continuous domain of the continuous variable $X_j$ that corresponds to the mean of  its discretized counterpart $\tilde{X}_j$.  We provide the detail in line 199-210 in our revised submission.
> >
> > > Q. Definition 3.4. Isn’t this simply the formula for the sample covariance?
> >
> > **A:** You are absolutely correct. Our intention was to unify all three cases (continuous, discretized, and mixed) using the bridge equation. However, your feedback made us realize that including the bridge equation for continuous pairs is redundant. We have removed this unnecessary case in the revised version. The bridge equation now specifically addresses scenarios involving discretized observations. Further details are provided in lines 257–262 of the revised submission.
> >
> > >Q. Theorem 3.5. I am confused by what is the rigorous statement in this theorem?
> >
> > **A:** Thank you for your question, which has helped us improve the clarity of our paper. You are absolutely correct—we intended to state that the statistic is asymptotically normal as $n$  increases. We have revised the expression to make this clear in the updated version. The details can be found in lines 293–297 of the revised submission.
> >
> > > Q. Theorem 3.9. Where is the $\hat{\beta}_{jk}$ formally defined?
> >
> > Thank you for sharing the content. Here's a refined version of your response:
> >
> > Certainly! Here's the revised text with properly formatted LaTeX expressions displayed normally:
> >
> > A: Sorry for the confusion.
> >
> > The $\hat{\beta}\_{jk}$  is defined in Equation 14 of the revised submission Equation 13 in the original submission and serves as the estimation of $\beta\_{jk}$.  The $\hat{\beta}\_{j,k}$ can be directly obtained since $\hat{\mathbf{\Sigma}}$ can be computed using observations only.
> >
> > > Q. I suggest to test several established datasets in BN structure learning such as ASIA, CHILD, WATER, and ANDES.
> >
> > A: Thanks for your great advice. We have checked the dataset and found out it's kind of misaligned with the motivation of our paper. Most of those discrete variables are inherently discrete rather than inherently continuous with discretized observations. For example, the dataset ASIA contains variables like "if the patient smokes", "if the patient has cancer".  To reduce your concerns, we will further investigate these datasets and select appropriate variables for the experiments. Once we have identified them, we will let you know the results as soon as possible.

---

> > > ### Author Response · Authors · 2024-11-25
> > > **Point-to-Point Rebuttal**
> > >
> > > Q.  Similar to the other experiment section, here, non-Gaussian distributions should be tested for structure learning.
> > >
> > > A: Thanks for your question. We kindly point out that the experiments on non-Gaussian are already conducted. As demonstrated in **Appendix E.1 of the revised version (Appendix C.1 of the original submission)**  and **Figures 5 and 6** where we present causal discovery results for various distributions including linear uniform, linear student distribution, linear exponential, and nonlinear Gaussian distributions. From the experiment, DCT still shows superior performance than other baselines.
> > >
> > >
> > > > Q.  This is a very interesting idea to use a DAG to invest the discretization problem; however, the authors implicitly assume that the DAG is faithful.
> > >
> > > Thank you for your really insightful comments. We would like to share that formulating certain common problems as a DAG is actually a well-established practice. Examples include handling missing values [1], measurement errors [2], and deterministic relationships [3]. You are absolutely correct that we assume the DAG is faithful, which may be somewhat restrictive. However, we also aim for Theorem 2.1 to emphasize the impact of discretization and highlight the urgency of addressing this issue.
> > >
> > >
> > >
> > > > Q. So all discretized variables are Gaussian? This is a limitation.
> > >
> > > Thank you for your great question. As stated in Appendix B9.2 (original section 3.4), we acknowledge that Gaussian assumption will limit the generality. However, we would like to share its rationale:
> > >
> > > 1. **Challenges in Conditional Independence**: Inferring the conditional independence of latent variables based on their discretized values is indeed a complex problem. The discretization drastically reduces available information. Without mild assumptions, establishing the statistics that reflect the real conditional independence and inferring its null distribution by only using those discretized values is particularly challenging and could even be overly ambitious.  In this paper, we rely on the property that (1). Gaussian assumption allows us to obtain ca onsistent estimator of latent covariance based on discretized observations thanks to its parametric form. (2) The Gaussian model has the property to infer conditional independence based on the covariance only.
> > > 2. **Empirical  Performance**: Although the theory requires that the latent continuous variables follow a multivariate Gaussian distribution, we empirically validate the effectiveness of DCT even when the assumptions are violated. As demonstrated in **Appendix E.1 of the revised version (Appendix C.1 of the original submission)**  and **Figures 5 and 6** where we present causal discovery results for various distributions including linear uniform, linear student distribution, linear exponential, and nonlinear Gaussian distributions. From the experiment, DCT still shows superior performance than other baselines.
> > > 3. **Popularity of Copula Model**:  The assumption of multivariate Gaussian, also called Gaussian copula model, is well-studied and widely accepted in the community. There is a substantial body of work demonstrating the effectiveness of the copula model in various scenarios [4] [ 5].
> > > On the other hand, we hope this paper can inspire the community to propose a more general and powerful solution to handle this obvious but overlooked spurious conditional dependence caused by discretization.
> > >
> > >
> > > > Was DCT ever defined? What does it stand for?
> > >
> > > Thanks for your question. DCT stands for **Discretization-Aware Conditional Independence Test.**
> > >
> > >
> > >
> > > [1] Tu R, Zhang C, Ackermann P, et al. Causal discovery in the presence of missing data[C]//The 22nd International Conference on Artificial Intelligence and Statistics. Pmlr, 2019: 1762-1770.
> > >
> > > [2] Zhang K, Gong M, Ramsey J, et al. Causal discovery in the presence of measurement error: Identifiability conditions[J]. arXiv preprint arXiv:1706.03768, 2017.
> > >
> > > [3] Li L, Dai H, Al Ghothani H, et al. On Causal Discovery in the Presence of Deterministic Relations[C]//The Thirty-eighth Annual Conference on Neural Information Processing Systems.
> > >
> > > [4] Fan, J., Liu, H., Ning, Y., and Zou, H. High dimensional  semiparametric latent graphical model for mixed data.   Journal of the Royal Statistical Society Series B: Statistical Methodology, 79(2):405–421, 2017.
> > >
> > > [5] Zhang A, Fang J, Hu W, et al. A latent Gaussian copula model for mixed data analysis in brain imaging genetics[J]. IEEE/ACM transactions on computational biology and bioinformatics, 2019, 18(4): 1350-1360.

---

> > > > ### Author Response · Authors · 2024-11-27
> > > > **Did we address your concerns properly?**
> > > >
> > > > Dear Reviewer i1w6,
> > > >
> > > > Thank you so much for dedicating your time and effort to reviewing this submission.  We are wondering whether your concerns have been properly addressed. If you have further comments, we hope for the opportunity to respond to them.  We are looking forward to your feedback.
> > > >
> > > > Best regards,
> > > >
> > > > Authors of submission #6671

---

> > > > ### Author Response · Authors · 2024-12-01
> > > >
> > > > Dear Reviewer i1w6,
> > > >
> > > > Thank you for having taken the time to provide us with your valuable comments, which helped improve our paper. This is a gentle reminder that the discussion period is nearing its conclusion, we hope you have taken the time to consider our responses to your review. If you have any additional questions or concerns, please let us know so we can resolve them before the discussion period concludes. If you feel our responses have satisfactorily addressed your concerns, it would be greatly appreciated if you could raise your score to show that the existing concerns have been addressed.
> > > >
> > > > Thank you!
> > > >
> > > > The Authors

---

> > > > > ### Comment · Reviewer_i1w6 · 2024-12-02
> > > > >
> > > > > Thanks for your revisions and responses. I think still the paper is a bit rushed and needs polishing to avoid redundancy and improve clarity. For example, now 133 and 150 say the same thing about the use of hat for the notations. Some examples can help as well.
> > > > >
> > > > > As with the DAGs I mentioned such as ASIA, the point is to use the DAG structure. For the parameters, you may use whatever fits your framework, e.g., a linear Gaussian distribution.
> > > > >
> > > > > My main concern though is what I stated in my Weakness 2, i.e., lack of proper experiments for the application of the test to structure learning. What the authors have now are only for [extremely] sparse DAGs. Therefore, the paper seems to better fit a statistical venue. I keep my original score.
> > > > >
> > > > > PS. Some parts seemed to be written by AI, although there's perhaps nothing wrong with it and I tried not to take this into account:
> > > > > > Thank you for sharing the content. Here's a refined version of your response:
> > > > >
> > > > > > Certainly! Here's the revised text with properly formatted LaTeX expressions displayed normally:

---

> ### Author Response · Authors · 2024-12-03
>
> Thank you for the insightful discussion, which has helped us better understand your concerns. Out of respect for the reviewers, we only used GPT to check the grammar.
>
> Indeed, the experiment included in the main results (Section 4 in the revised version) investigates only sparse DAGs, while additional experiments with denser graphs are provided in Appendix E.2. However, we would like to emphasize that, as discussed in the Appendix, the primary focus of DCT is on cases where conditional independence (CI) is incorrectly judged as conditional dependence due to discretization. With an increase in edges, such cases of true CI become rarer,  diminishing any advantages that DCT might have compared to other tests. Therefore, **denser graphs are not the primary focus of our paper.**
>
> To alleviate your concerns, given the tight timeline, we conducted some experiments on denser graphs. Here are the results where the number of nodes $p=10$ and there are in total $2p$ edges. We use $n=500$ samples:
>
> |    Skeleton        | F1           | Precision  | Recall     | SHD   |
> | ---------- | ------------ | ---------- | ---------- | ----- |
> | DCT        | 0.5398       | 0.76000000       | 0.41666667      | 13    |
> | Fisherz    | 0.57156863 | 0.58884804 | 0.55555556 | 15    |
> | Chi-square | 0.5400035    | 0.57685574 | 0.50925926 |  15.5 |
>
> We can see that the precision and SHD of DCT are still better than other baselines, while the recall is lower due to the fact that discretization tends to keep all nodes connected. The experiment of other settings investigating denser graphs is going on and we will let you know the result once it is finished.
>
> Regardless, we **sincerely** **thank** the reviewer for the dedicated review and insightful questions, which made us realize the **shortcomings** in the readability of the paper in the first submission.  We have made every effort to improve the clarity and quality of the paper in its revised version.

---

### Official Review · Reviewer_WY1e · 2024-10-28

**Soundness:** 3
**Presentation:** 3
**Contribution:** 3
**Rating:** 6
**Confidence:** 3

**Summary:**

In this paper, a CI test has been designed specifically for handling the presence of discretization. An appropriate test statistic for the CI of latent continuous variables, based solely on discretized observations, has been derived. The key is to build connections between the discretized observations and the parameters needed for testing the CI of the latent continuous variables. To achieve this, first, the bridge equations have been developed to estimate the covariance of the underlying continuous variables with discretized observations. Then, a node-wise regression has been applied to derive appropriate test statistics for CI relationships from the estimated covariance. By assuming that the continuous variables follow a Gaussian distribution, it has been derived the asymptotic distributions of the test statistics under the null hypothesis of CI.

**Strengths:**

1- Improving CI test accuracy is an essential and practical problem, particularly for structure learning in Bayesian networks and causal discovery.

2- The proposed method is novel, and the solution provided is relatively simple and efficient.

**Weaknesses:**

1- In (1), the discretization model (function) is unclear. How many discretization levels are used in this formulation? In addition, While the main idea is well presented, the authors have not clearly conveyed the details of the proposed method.

2- Providing a step-by-step explanation of CDT in a numerical example would enhance understanding of the proposed method.

3- The proposed method has improved the accuracy of the CI tests. However, another significant challenge with CI tests is the computational burden, particularly when there are many conditioning variables and a large sample size. Including a complexity analysis or a runtime comparison with existing methods in the numerical results would help demonstrate the impact of the proposed method.

4- Does not the number of discretizing levels affect the accuracy (Type I and Type II error) of the DCT method? In Figure 2 the effect of the number of discretizing levels on Fisherz and Chi2 methods has been checked and shown but the impact of it on CDT has not been reported. Why? I think the number of discretizing levels affects the accuracy of the CDT.

5- Using practical examples for the application of the CDT in causal discovery can help to indicate the effectiveness of the proposed method in practical and real problems.

6- In proof of Theorem 2.1, in appendix A.1, “Var(X2| \tilde{X2}) will never be zero since  \tilde{X2} is the discretized version of X2 (information loss).” is intuitively correct but it is not a precise mathematical expression for proof. Is there any reference or more precise explanation for that?

7- According to Figures 5, 6, and 8, CDT outperforms other methods in F1 score and Precision but generally performs worse in Recall.

**Questions:**

1- In the numerical results for the causal discovery example, In Figure 3, with increasing the number of samples F1 score, and precision are reduced for Fisher_z and Chis-q. Usually, increasing the number of samples increases accuracy. What is your explanation?

2-  The proposed CI test was developed and analytically proven for the Gaussian distribution. Although it has been applied to other distributions in the numerical results, can the proposed method be analytically extended to other continuous distributions or to handle nonlinear relationships?

---

> ### Author Response · Authors · 2024-11-25
> **Point-to-Point Rebuttal**
>
> > Q. The discretization model (function) is unclear. How many discretization levels are used in this formulation?
>
> A: Thanks for your question. We kindly note that we claimed in line108-109: "  The cardinality of the domain after discretization is at least 2 and smaller than infinity."
>
> > Q:  Providing a step-by-step explanation of CDT in a numerical example would enhance understanding of the proposed method.
>
> A: We sincerely thank the reviewer for the valuable feedback. We have included the Pseudo code of DCT  in Appendix~D in the revised version to enhance the clarity of the paper.
>
> > Q: The proposed method has improved the accuracy of the CI tests. However, another significant challenge with CI tests is the computational burden, particularly when there are many conditioning variables and a large sample size.
>
> A: While our proposed test exhibits higher computational complexity compared to traditional methods like the Fisher's z-test and the Chi-square test, we believe the benefits justify this trade-off. The main computational challenge arises in calculating the variance of the null distribution as outlined in Theorem 3.8.
>
> Here's a breakdown of the computational steps:
>
> 1. **Compute Inner Products**: For each $i$ from 1 to $ n $, we compute the inner product $\mathbf{a}[k]^\top \text{vec}(\mathbf{B}^i)$.
>    - **Cost per $ i $**: This operation requires $ O(p^2) $ computations because $ p(p-1) \approx p^2 $ for large $ p $.
>
> 2. **Sum the Squares**: We then sum all the computed inner product values.
>    - **Total Cost**: This step has a computational cost of $ O(n)$.
>
> The total computational cost amounts to $ O(np^2) $.
>
> At the same time, our current method is well-suited for parallelization, as the calculations for estimating the variance in both unconditional and conditional independence tests are parameter-free and can be efficiently distributed across multiple processors.
>
>
> > Q.  Does not the number of discretizing levels affect the accuracy (Type I and Type II error) of the DCT method?
>
> Thank you for highlighting the confusion. We would like to take this opportunity to clarify our approach. As explained in Section 3.1.1 of the revised version ( lines 174–180 of the original submission), to unify both discretized pairs and mixed pairs, we treat every discretized variable as a binary variable. Consequently, the level of discretization does not affect the performance of DCT.
>
>
> > Q. Using practical examples for the application of the CDT in causal discovery can help to indicate the effectiveness of the proposed method in practical and real problems.
>
> We kindly note that the real-world experiment is already included in **Appendix D.4 of the revised version (Appendix C.4 of the original submission).** In this experiment, we perform causal discovery on the Big Five dataset, where each variable represents the level of agreement to a specific question with discrete choices. This case directly aligns with the causal graph introduced in the paper. Specifically, the variable "I worry about things" is conditional independent with "I seldom feel blue" given " I often feel blue". However, given their discretized observations, traditional tests fail to capture the true CI relationships while DCT successfully identifies them.
>
> > Q. Var(X2| \tilde{X2}) will never be zero since \tilde{X2} is the discretized version of X2 (information loss).
>
> Thanks for your great question. We have included more detail in our revised version to clarify this point. The basic idea is as follows: The conditional variance $Var(X_2| \tilde{X}_2$ can be zero if and only if the $X_2$ is determined exactly by $\tilde{X}_2$ without any randomness, which clearly doesn't hold true in the presence of discretization.
>
>
> > Q. According to Figures 5, 6, and 8, CDT outperforms other methods in F1 score and Precision but generally performs worse in Recall.
>
> Thank you for your question. We would like to clarify that this phenomenon is already explained in **lines 528–530 of the revised version (lines 526–529 in the original submission)**. In essence, other tests tend to infer **conditional dependence** in most cases due to the effects of **discretization**, resulting in nearly all nodes being connected.
>
> To elaborate, recall is calculated as:
>
> $\text{Recall} = \frac{\text{TP}}{\text{TP} + \text{FN}},$
>
> where:
> - **True Positives (TP):** The number of edges correctly identified by the algorithm as existing in the ground truth graph.
> - **False Negatives (FN):** The number of edges in the ground truth graph that the algorithm failed to identify.
>
> In scenarios where the algorithm connects most nodes (traditional tests due to discretization), such as in the extreme case of a fully connected graph, the number of False Negatives (FN) approaches zero. As a result, the recall approaches one, regardless of whether the inferred graph accurately reflects the true causal relationships.

---

> > ### Author Response · Authors · 2024-11-25
> > **Point-to-Point rebuttal**
> >
> > > Q.  In the numerical results for the causal discovery example, In Figure 3, with increasing the number of samples F1 score, and precision are reduced for Fisher_z and Chis-q. Usually, increasing the number of samples increases accuracy. What is your explanation?
> >
> > Thanks for the question and we would like to take this chance to highlight our contribution.  **For all tests without awareness of discretization, more samples will only lead to more confident wrong conclusions**.
> >
> > As our key motivation and illustrated in the causal graph of Figure~1, the discretization can lead to the wrong conclusion of CI. This is because the traditional tests are actually measuring the CI among discretized observations if $\tilde{X}_1$ conditional independent with $\tilde{X}_2$ given $\tilde{X}_3$, rather than the real interested continuous counterparts $X_1$ conditional independent with $X_3$ given $X_2$.
> >
> > For tests without correction, increasing the sample size improves their ability to detect relationships within the discretized variables. This makes them more likely to correctly conclude that $\tilde{X}_1 \not \perp \tilde{X}_3 | \tilde{X}_2$, which, however, is the opposite of the true relationship among the continuous variables of interest, $X_1 \not \perp X_3 | X_2$
> >
> >
> > > Can the proposed method be analytically extended to other continuous distributions or to handle nonlinear relationships?
> >
> > We appreciate the reviewer’s insightful question. The answer is **no** under the current framework. Inferring the conditional independence (CI) relationship of the original continuous variables based on their discretized observations requires two essential components:
> >
> > 1. Statistics of the continuous variables derived solely from the discretized observations.
> > 2. These statistics must accurately reflect the CI relationship.
> >
> > Using a parametric form, the first component is straightforward to obtain—for example, we can estimate the correlation of the original continuous variables based on discretized observations. However, without assuming a Gaussian distribution, correlation does not equate to conditional independence. This highlights the necessity of our linear Gaussian assumption.
> >
> > For nonlinear relationships, the problem becomes more challenging. Nonlinear dependencies typically involve nonparametric approaches, and discretization drastically reduces the available information. Without appropriate assumptions, even deriving meaningful statistics is difficult, let alone conducting inference on CI relationships.

---

> ### Author Response · Authors · 2024-11-27
> **Concerns properly addressed?**
>
> Dear Reviewer WY1e,
>
> Thanks for the time and effort you dedicated to reviewing our submission.  We hope your concerns have been properly addressed, and if you have further comments, we hope for the opportunity to respond to them.  Your feedback would be appreciated.
>
> Best wishes,
>
> Authors of submission #6671

---

> ### Comment · Reviewer_WY1e · 2024-12-01
>
> Thank you to the authors for their clear and comprehensive responses. Most of my major questions have been addressed, and the paper has been revised effectively. Regarding my comment on applying the proposed method to causal discovery, I was referring to using real-world structures like ASIA, SACHS, or ALARM. Based on the responses and modifications, I have updated my score from 3 to 6.

---

> > ### Author Response · Authors · 2024-12-01
> >
> > We sincerely appreciate your thoughtful review and for raising your score upon re-evaluating our submission. Your constructive feedback has greatly improved our paper. Thank you for taking the time to carefully consider our revisions and for acknowledging the contributions of our research.

---

### Official Review · Reviewer_nGxm · 2024-11-03

**Soundness:** 3
**Presentation:** 2
**Contribution:** 3
**Rating:** 5
**Confidence:** 3

**Summary:**

This paper addresses a specific challenge in confidence independence (CI) testing when variables of interest are discretized. In practice, accurately measuring continuous variables is often difficult due to data collection limitations, resulting in a common reliance on discretized values for these variables. To tackle this, the authors propose a unified bridge equation to estimate the covariance between two random variables by binarizing them. They then employ nodewise regression to recover precision coefficients that capture the conditional dependence among latent continuous variables. An appropriate test statistic is proposed, and its asymptotic distribution under the null hypothesis is derived.

**Strengths:**

The paper presents a well-established theoretical foundation and methodology. The authors begin with a straightforward theorem to illustrate the impact of discretization, emphasizing the potential need for correction in CI testing. The theoretical framework is clearly articulated and thorough.

**Weaknesses:**

The paper needs improvement in writing. There are several instances where the notations are unclear, inconsistent, and occasionally used imprecisely (see Questions for more details).

**Questions:**

Major questions:
1. The authors empirically verify the Type I and Type II errors of the test. I’m curious if the asymptotic Type I error and power of the test could also be theoretically verified as well. I'm assuming the asymptotic power or Type I error will depend on $\mathbf{Λ}$ as well.
2. Have the authors considered providing a quantitative rate theory to support the test’s validity for smaller datasets? Additionally, are there any debiasing corrections that could be applied for small dataset sizes?
3. [Optional] The authors present a comprehensive framework based on the assumption that the random vector follows a multivariate normal distribution. I wonder if they have any thoughts on extending this framework to accommodate non-normal variables.

Minor questions:
1. Line 34: undefined notation $X_{j_1}$. More explanation on what ${j_1}$ means or may consider using $X_{1}$ and $X_{2}$ here.
2. Line 89-90: same issue as in 1.
3. Line 122: inconsistent notations $\mathrel{\text{{$\perp\mkern-10mu\perp$}}}$ versus $\bot$ defined at Line 36. The former notation is more commonly referred as conditional independence in literature.
4. Line 126: what does DCT stand for? Should introduce the abbreviation beforehand.
5. Line 754: minor typo, should be $X_1\mathrel{\text{{$\perp\mkern-10mu\perp$}}}X_3 | X_2$
6. Line 137: undefined subscript $\sigma_{j_1, j_2}$. Consider using different subscripts for $\sigma_{j_1, j_2}$ and $\omega_{k,l}$ to distinguish them.
7. Line 362: remove ) after $vec(B^i_{-j})^T$.
8. Line 369: as much as I love the theory, I do find the mathematical notations concerning. I don't really see people using $vec$ to denote the rows of a matrix.

---

> ### Author Response · Authors · 2024-11-25
>
> We sincerely thank the reviewer for the **comprehensive and insightful** feedback, which highlighted areas in our original submission that lacked clarity and may have led to confusion. In response, we have made the following key revisions to improve the organization and readability of our paper:
>
> 1. **Notation Table**: We have included a complete table of notations in Appendix A and provided definitions for key terms at the beginning of Section 3 to improve accessibility and understanding. The notations now are consistent and simplified.
>
> 2. **Algorithm Pseudocode**: To further enhance clarity, we have added the pseudocode for our core algorithm, DCT, in Appendix D. This addition provides a more straightforward explanation of the algorithm’s workflow and implementation.
>
> 3. **Reorganization and Intuition**: We have reorganized the sections discussing the **bridge equation**, **calculation of covariance**, and **independence tests**. Where necessary, we have introduced new notations and supplemented them with intuitive explanations to complement the formal mathematical definitions.
>
> These changes aim to address the concerns raised and improve the overall clarity of our paper.

---

> > ### Author Response · Authors · 2024-11-25
> >
> > Q. I’m curious if the asymptotic Type I error and power of the test could also be theoretically verified as well. I'm assuming the asymptotic power or Type I error will depend on $\Lambda$ as well.
> >
> > A: Thanks for your great question. We apologize for not fully understanding your reference to "theoretically verified." Based on our current interpretation, we believe you may be referring to the distributions of $\hat{\sigma}\_{j\_1j\_2}−{\sigma}\_{j\_1j\_2}$ and $\hat{\beta}\_{j,k} - \beta\_{j,k}$. If so, the two main theorems in our paper explicitly describe these distributions. To test the power, we can simply assign specific values to the true parameters $\beta\_{j,k}$ and $\sigma\_{j\_1j\_2}$. However, if you meant something else, we would greatly appreciate further clarification and would be happy to address your concerns.
> > You are totally right since the $\Lambda$ directly influences the variance of distribution of $\sigma\_{j\_1j\_2} - \sigma\_{j\_1j\_2}$, which is directly related to $\hat{\beta}\_{j,k} - \beta\_{j,k}$.
> >
> >
> > >Q. Have the authors considered providing a quantitative rate theory to support the test’s validity for smaller datasets? Additionally, are there any debiasing corrections that could be applied for small dataset sizes?
> >
> > Thank you for your insightful question. Both the independence test and conditional independence (CI) test in our framework rely on the Central Limit Theorem, which guarantees convergence of statistical inference at a rate of $1/\sqrt{n}$.
> >
> > A straightforward approach to mitigate the small dataset bias is to apply naive methods like bootstrap resampling. We greatly appreciate your suggestion and will further explore quantitative rate theories and debiasing corrections to address the challenges associated with small sample sizes in future work.
> >
> > > Q. I wonder if they have any thoughts on extending this framework to accommodate non-normal variables.
> >
> > Thank you for your insightful question. At present, the answer is no. Inferring the conditional independence (CI) relationship among original continuous variables from their discretized observations requires three ingredients:
> >
> > 1. Statistical measures of the continuous variables derived solely from the discretized observations.
> > 2. The ability of these measures to accurately represent the CI relationship.
> > 3. The inference of these measures.
> >
> > The first component is relatively straightforward under a parametric framework—for instance, we can estimate the correlation of the original continuous variables. However, correlation alone does not imply conditional independence unless we assume a Gaussian distribution. Furthermore, discretization significantly reduces the available information, making the construction of meaningful measures challenging. Even if such measures are constructed, inference remains nontrivial without additional assumptions.

---

> ### Author Response · Authors · 2024-12-01
>
> Dear Reviewer nGxm,
>
> Thank you for having taken the time to provide us with your valuable comments, which helped improve our paper. This is a gentle reminder that the discussion period is nearing its conclusion, we hope you have taken the time to consider our responses to your review. If you have any additional questions or concerns, please let us know so we can resolve them before the discussion period concludes. If you feel our responses have satisfactorily addressed your concerns, it would be greatly appreciated if you could raise your score to show that the existing concerns have been addressed.
>
> Thank you!
>
> The Authors

---

### Official Review · Reviewer_mvCf · 2024-11-03

**Soundness:** 4
**Presentation:** 4
**Contribution:** 3
**Rating:** 8
**Confidence:** 4

**Summary:**

This paper tackles the challenge of testing conditional independence (CI) with a focus on accommodating discretized data, a crucial aspect for various applications. The authors introduce bridge equations to estimate covariance and establish asymptotic normality, facilitating an unconditional independence test. For the conditional independence test, they employ nodewise regression to recover precision coefficients. The effectiveness of the proposed method is substantiated through rigorous theoretical analysis and empirical validation.

**Strengths:**

This paper is well-written and easy to follow. The proposed CI test effectively corrects and infers the true CI relationships among latent continuous variables, recognizing that discretized variables may show different conditional independence compared to their continuous counterparts.

**Weaknesses:**

The authors primarily discretized variables into binary categories, which could potentially lead to some efficiency loss.

**Questions:**

1. In Theorem 2.1, the authors emphasize the importance of establishing correct CI relationships among the latent continuous variables of interest. However, if we can only observe $\tilde X_2$, why is it necessary to infer the original conditional relationships? Could the authors provide practical examples to clarify this necessity?

2. In line 192, multiple definitions for $\tilde h_j$ are possible, such as substituting $E\tilde X_j$ with other quantities. Considering that $\tilde h_j$ impacts the asymptotic variance of the estimated covariance as noted in Theorem 3.5, is there an optimal choice for $\tilde h_j$?

---

> ### Author Response · Authors · 2024-11-25
>
> >Q. The authors primarily discretized variables into binary categories, which could potentially lead to some efficiency loss
>
> A. We are grateful for the reviewer’s insightful observation. You are totally right. As we discussed in Appendix~H, in the current framework, the estimation of hidden covariance $\sigma_{j_1,j_2}$ is accomplished by solving a single bridge equation. This equation, can be interpreted as looking for the suitable $\sigma_{j_1,j_2}$ to match the "region" in continuous domains, linking the covariance with discretized observations. However, there might be multiple bridge equations available while the current framework only allows the usage of one of them. This is, as discussed in the paper, by far the largest limitation. In the future, we will try to make efforts towards this problem to solve more bridge equations thus improving the sample efficiency.
>
>
> >Q. If we can only observe $\tilde{X}_2$, why is it necessary to infer the original conditional relationships?
>
> A. Thanks for your great question and we would like to take this chance to highlight our motivation of this paper. Traditionally, when we talk about **CI test for discrete variables, we take it for granted that those variables are inherently discrete** (variables like gender). However, numerous discrete variables, are inherently continuous but only discretized observations available. Just like the real-world experiment we have done in this paper, the variables representing the agreement to a certain question is categorized into five levels.
> In this case, we care about the CI relations among the original continuous variables rather than their discretized observations. However, as illustrated in the causal graph, this discretization leads to the possible failure conclusion of traditional tests.
>
> > Q. Could the authors provide practical examples to clarify this necessity?
>
> A: One simple example could be the real-world experiment we included in this paper. The variable "I worry about things" is conditional independent with "I seldom feel blue" given " I often feel blue". However, given their discretized observations, traditional tests fail to capture the true CI relationships while DCT successfully identifies them.
>
> >Q. Multiple definitions for $h_j$ are possible, such as substituting $E[X_j]$ with other quantities.
>
> A: Thanks for your insightful question.  You are totally right that there are multiple ways to define $h_j$. We choose the mean of $E \tilde{X}_j$ as the quantity just for its simplicity. Currently, there is no theoretical progress to systematically determine the optimal $h_j$. However, we believe it is doable empirically. One naive solution would be to test $h_j$ with different quantiles ($E \tilde{X}_j$,  $2E \tilde{X}_j$, etc.), calculate the corresponding variance using Theorem~3.3 and choose the one with the smallest variance.

---

### Official Review · Reviewer_mKQu · 2024-11-04

**Soundness:** 3
**Presentation:** 3
**Contribution:** 3
**Rating:** 6
**Confidence:** 3

**Summary:**

The authors examine the problem of developing conditional independence (CI) tests when a subset of variables is discretized. This discretization introduces bias into existing CI test methods. They apply the concept of bridge equations to accommodate discretized random variables. Additionally, they demonstrate that the asymptotic distribution of their test under the null hypothesis (conditional independence) is normal and finally evaluate their test empirically.

**Strengths:**

The problem studied is highly relevant and significant across various fields.
The theoretical findings and asymptotic analysis provide valuable insights, and the empirical results are promising.

**Weaknesses:**

The method is developed under the assumption of Gaussianity, which is quite limiting. Although the authors argue that this assumption is not restrictive, I find their reasoning unconvincing. The analysis relies on correlation measures (covariance and precision matrices), which only reveal dependency and independence structures in the case of multivariate normal distributions.

Additionally, Equation (5) lacks precision, as a Taylor expansion includes a residual term. Specifically, the last term should be $f'(\theta_*)$  for a $\theta_*$ between $\theta_0$ and $\hat{\theta}$ rather than $f'(\theta_0)$.

The empirical results are limited for showing the generality of the developed method for non-gaussian settings.

**Questions:**

Please see the above comments.

---

> ### Author Response · Authors · 2024-11-25
> **Point-to-point rebuttal**
>
> > Q: The method is developed under the assumption of Gaussianity, which is quite limiting.
>
> A: Thank you for your insightful feedback. We acknowledge that the assumption of multivariate Gaussian distributions can limit the generality of the proposed test. However, we would like to share a few points regarding its reasonability:
> 1. **Challenges in Conditional Independence**: Inferring the conditional independence of latent variables based on their discretized values is indeed a complex problem. The discretization drastically reduces available information. Without mild assumptions, establishing the statistics that reflect the real conditional independence and inferring its null distribution by only using those discretized values is particularly challenging and could even be overly ambitious.  In this paper, we rely on the property that (1). Gaussian assumption allows us to obtain a consistent estimator of latent covariance based on discretized observations thanks to its parametric form. (2) Gaussian model has the property to infer conditional independence based on the covariance only.
> 2. **Empirical  Performance**: Although the theory requires that the latent continuous variables follow a multivariate Gaussian distribution, we empirically validate the effectiveness of DCT even when the assumptions are violated. As demonstrated in **Appendix E.1 of the revised version (Appendix C.1 of the original submission)**  and **Figures 5 and 6** where we present causal discovery results for various distributions including linear uniform, linear student distribution, linear exponential, and nonlinear Gaussian distributions. From the experiment, DCT still shows superior performance than other baselines.
> 3. **Popularity of Copula Model**:  The assumption of multivariate Gaussian, also called Gaussian copula model, is well-studied and widely accepted in the community. There is a substantial body of work demonstrating the effectiveness of the copula model in various scenarios [1] [2].
> On the other hand, we hope this paper can inspire the community to propose more general and powerful solution to handle this obvious but overlooked spurious conditional dependence caused by discretization.
>
>
>
> > Q. Additionally, Equation (5) lacks precision, as a Taylor expansion includes a residual term.
>
> Thanks for raising the looseness. You are totally right we omitted the second-order terms and more. We have clarified this point in our revised submission.
>
> > Q. The empirical results are limited for showing the generality of the developed method for non-gaussian settings.
>
> Thanks for your question. We kindly point out that the empirical results validating the robustness of DCT in non-Gaussian settings are already listed in **Appendix E.1 of the revised version (Appendix C.1 of the original submission)** and **Figures 5 and 6** where we present causal discovery results for various distributions including linear uniform, linear student distribution, linear exponential, and nonlinear Gaussian distributions. The experiments demonstrate the superiority of DCT even when assumptions are violated.
>
>
> [1] Fan, J., Liu, H., Ning, Y., and Zou, H. High dimensional  semiparametric latent graphical model for mixed data.   Journal of the Royal Statistical Society Series B: Statistical Methodology, 79(2):405–421, 2017.
>
> [2] Zhang A, Fang J, Hu W, et al. A latent Gaussian copula model for mixed data analysis in brain imaging genetics[J]. IEEE/ACM transactions on computational biology and bioinformatics, 2019, 18(4): 1350-1360.

---

### Meta-Review · Area_Chair_haCz · 2024-12-17

**Metareview:**

Since there is a large span for the ratings of the reviewers (but 4 out of 5 judge the content "good"), I have read the paper and went through the reviews, the rebuttal stage, the updates in the paper.

The key part in the negative opinion of i1w6 is the sparsity of DAG used. In response, the authors have performed a lightweight experiment on denser graphs, which shows quite unbalanced precision and recall compared to the other baselines (and a smaller F1). The key argument of the authors, however, is that denser graphs should not be taken as the focus of the paper. More than the additional experiments, I am happy to consider this argument. I also note that the authors have made a *very* substantial effort of writing / rewriting parts of the paper as a consequence on the rebuttal process, which made reviewer WY1e bump their score up by 3 points.

i1w6's suggestion to submit to a statistical journal is reasonable given the paper's content, but it does not prevent the paper to have merits for ICLR as well, and I do consider it has such merits, mainly because of the broad interest of the question tackled -- and not just for causal inference --.

In conclusion, I believe the authors have made a very substantial effort to answer the reviewers comments and can be taken into consideration for acceptance.

**Additional Comments On Reviewer Discussion:**

The discussions focused on the two reviewers who had initially a 3 rating (WY1e, i1w6). The authors managed to reverse the polarity of the rating of WY1e. The case of i1w6, while relying on fair and sound comments, was arguably harder to tackle -- it relied a lot on the problems relevant to the paper and the venue relevant for its publication. I also consider that the authors made a very decent job of explaining and arguing in favor of their work.

---

### Decision · Program_Chairs · 2025-01-22

Accept (Poster)